# The persistence of memory in ionic conduction probed by nonlinear optics

Andrey D. Poletayev[1,2,3 ✉], Matthias C. Hoffmann[4], James A. Dawson[5,6], Samuel W. Teitelbaum[1,7,9], Mariano Trigo[1,7], M. Saiful Islam[3,8] & Aaron M. Lindenberg[1,2,7 ✉]

Predicting practical rates of transport in condensed phases enables the rational design of materials, devices and processes. This is especially critical to developing low-carbon energy technologies such as rechargeable batteries[1–3]. For ionic conduction, the collective mechanisms[4,5], variation of conductivity with timescales[6–8] and confinement[9,10], and ambiguity in the phononic origin of translation[11,12], call for a direct probe of the fundamental steps of ionic diffusion: ion hops. However, such hops are rare-event large-amplitude translations, and are challenging to excite and detect. Here we use single-cycle terahertz pumps to impulsively trigger ionic hopping in battery solid electrolytes. This is visualized by an induced transient birefringence, enabling direct probing of anisotropy in ionic hopping on the picosecond timescale. The relaxation of the transient signal measures the decay of orientational memory, and the production of entropy in diffusion. We extend experimental results using in silico transient birefringence to identify vibrational attempt frequencies for ion hopping. Using nonlinear optical methods, we probe ion transport at its fastest limit, distinguish correlated conduction mechanisms from a true random walk at the atomic scale, and demonstrate the connection between activated transport and the thermodynamics of information.

Linking the mechanistic features of ion transport at the atomic level to the collective descriptors of macroscopic transport within a multiscale model, or as measured in a device, yields opportunities to design new processes and applications[3,9,13]. Fast-ion transport in the solid state commands particular attention due to its importance in energy and information technologies such as solid-state batteries and non-volatile memory[1–3]. Solid-state ion transport is composed of rapid translations of ions between lattice sites, called hops, which constitute the fundamental steps of diffusion and conduction (Fig. 1a).

However, the correspondence between atomistic and macroscopic regimes in ion transport is often not well characterized: whereas nanoscale ionic transport is heterogeneous, dispersive and non-ergodic[6,7], correlation-sensitive probes of atomistic paths analogous to single particle tracking in biophysics do not exist. Without such information, the maximum-entropy principle compels models of transport constructed from macroscopic measurements to assume that ion transport proceeds via a Markovian random walk[14,15]. From the standpoint of information theory[16], the full entropy of transport is evolved at every step of this memory-free process. Traditionally equated to hops in solid-state transport, these steps originate randomly with an attempt frequency $\nu_0$, and succeed with a probability determined by the Gibbs free energy of a transition state[6,11,12,17].

By contrast, models of correlated transport allow for the macroscopic process of conduction to consist of several interconnected steps. Examples include memory kernels in generalized master equations[18,19], Burnett-order nonlinear hydrodynamics[8,10] or kinetic competition[6,20]. In such models, the state of the material at time zero (Fig. 1b, dark green) influences transport dynamics over some non-negligible timescale (for example, $t_1$ in Fig. 1b). The information entropy of transport is expressed as a mutual entropy between configurations a time lag $t_1$ apart. This quantity increases (equivalently, mutual information decreases) until the full entropy of transport is produced, possibly over several successive atomistic steps or several consecutive correlated hops. Only at longer timescales (for example, at $t_2$ in Fig. 1b) can transport parameters such as the diffusion coefficient and ionic conductivity be constant-valued, as for a random-walk process. Indeed, picosecond to nanosecond studies of ion conductors[11,21,22] yield reduced activation energies, suggesting that the processes being probed at those frequencies may be incomplete with respect to overall conduction due to the persistence of such memory.

Understanding correlation effects in transport remains necessary to predict practical performance of ionic conductors from the atoms up[3,13], and to exploit nonlinear nanoscale transport phenomena[9] in devices. Here, we study the correspondence between the thermodynamics of ion transport and those of information[23,24]. Such a mechanistic investigation requires the ability to (1) impulsively trigger ion hops[3] on the short timescale of $1/\nu_0$, typically no more than 1 ps, and (2) track their outcomes over potentially much longer times[7,8]. Such

[1]Stanford Institute for Materials and Energy Sciences, SLAC National Laboratory, Menlo Park, CA, USA. [2]Department of Materials Science and Engineering, Stanford University, Stanford, CA, USA. [3]Department of Materials, University of Oxford, Oxford, UK. [4]Linac Coherent Light Source, SLAC National Accelerator Laboratory, Menlo Park, CA, USA. [5]Chemistry, School of Natural and Environmental Sciences, Newcastle University, Newcastle upon Tyne, UK. [6]Centre for Energy, Newcastle University, Newcastle upon Tyne, UK. [7]Stanford PULSE Institute, SLAC National Accelerator Laboratory, Menlo Park, CA, USA. [8]Department of Chemistry, University of Bath, Bath, UK. [9]Present address: Department of Physics, Arizona State University, Tempe, AZ, USA. ✉e-mail: andrey.poletayev@gmail.com; aaronl@stanford.edu

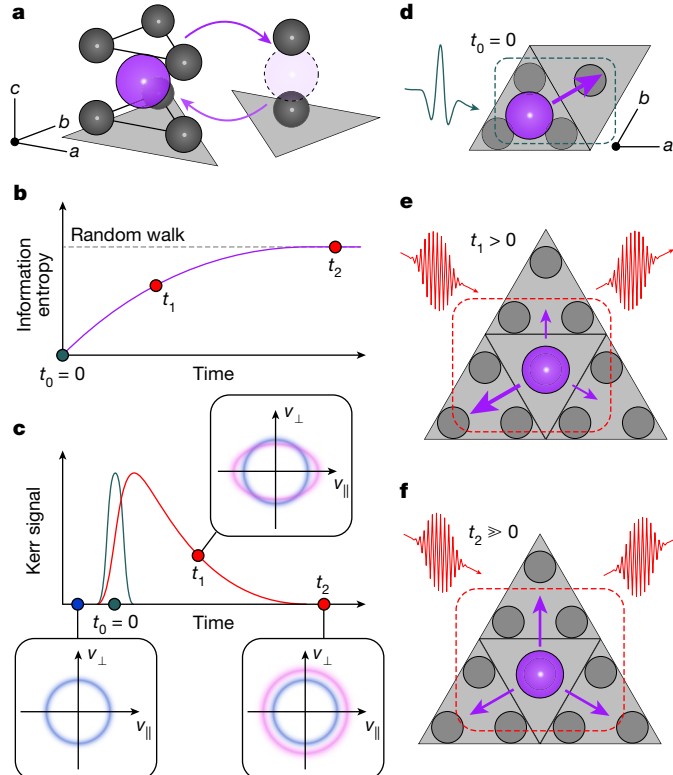

**Fig. 1 | Probing ionic transport in β-aluminas. a**, Mobile metal ions (purple) hop (arrows) between energetically and crystallographically distinct lattice sites (triangles) with two- and six-fold coordination by oxygen ions (black). The same lattice sites are depicted schematically in **d**–**f. b**, If at time $t_0 = 0$ (dark green) there is perfect knowledge of the state of the system, then at time $t_1 > 0$, some correlations persist. By time $t_2 \gg 0$, all correlations and memory are lost, and the full entropy of transport is produced as for a random walk (dotted line). **c**, Transient birefringence experiment in β-aluminas. **d**, At time $t_0 = 0$ (dark green), the terahertz pump triggers ion hops along its electric field vector. **e**, At time $t_1 > 0$, the subsequent hops retain a correlation with the impulse at $t_0$: back-hopping is predominant, which yields a birefringence signal (**c**, red). The anisotropic distribution of hopping directions at $t_1$ is probed as a contribution to the anisotropy in ionic velocities (**c**, pink). **f**, The pump energy fully thermalizes by time $t_2$ when the hopping directions reach isotropy.

a combination is inaccessible to current techniques[11], and even the vibrational nature of $v_0$ remains under debate[2,25,26]. Furthermore, since ion hops, unlike those of electronic carriers, are stochastic and lack clear spectral resonance signatures, single-pulse reciprocal-space or Fourier-transform probes are insufficient for probing them. Instead, correlations in nanoscale transport[7–10] could be probed using time-domain nonlinear optical pump–probe methods such as transient birefringence[27–29] by means of these methods' sensitivity to polarization. However, pump–probe studies of ionic dynamics have so far focused on coherent displacements of bound vibrational modes or order parameters[30,31], in which many ions undergo coherent sub-ångström motions, rather than rare large-amplitude displacements (typically 2–3 Å) such as ionic hops.

## Probing correlated hopping

We probe orientational correlations between ionic hops using a nonlinear optical method sensitive to ionic velocities (Fig. 1c). We trigger ionic hops with an impulsive single-optical-cycle pump (Fig. 1d) and measure the anisotropy in the directions of subsequent hops (Fig. 1e,f) as transient birefringence. The ionic response is pumped impulsively using single-cycle terahertz pulses with centre frequency near 0.7 THz

(refs. 32,33), and transient birefringence is probed in transmission mode (Extended Data Fig. 1). The general solution for electric fields $E$ in the sample at position $z$ along the pulse paths at pump–probe delay $t_1$ is[34,35]

$$E_{\text{probe}}(z, t) - E(z, t) \propto \frac{\partial}{\partial t}(P(z, t + t_1)E_{\text{probe}}(z, t)) \quad (1)$$

Here, the polarization $P$ arises from coherent displacements of vibrational modes or bound dipoles $Q_i$, plus an extra component from the history of hopping rates $H$:

$$P(t + t_1) = \sum_i Q_i(t + t_1) + \int_{-\infty}^{t+t_1} H(\tau)d\tau \quad (2)$$

The integral is zero until the pump drives ionic hops in a preferential direction (Fig. 1d). Because the emitted signal at $t_1$ is proportional to the time derivative of the polarization, it arises from ionic velocities, to which hopping rates $H(t_1)$ contribute. In the terahertz-pumped Kerr effect (TKE) geometry, the anisotropy of velocities and hopping rates is measured specifically (Fig. 1c). This anisotropy corresponds to a time-dependent preference for hopping along the direction defined by the terahertz pump relative to an orthogonal direction. The relaxation of hopping anisotropy corresponds to the loss of memory of the pump-driven impulse by the conducting ions during the temperature-activated solid-state diffusion process. If conduction occurs via a random walk, then hopping immediately returns to isotropy and no relaxation should be observable. Furthermore, the modes $Q_i$ that couple appreciably to $H$ contribute to attempt frequencies $v_0$.

## Picosecond hopping dynamics in β-aluminas

We first use the fast ionic conductors β-aluminas ($M_{1+2x}Al_{11}O_{17+x}$, where $x$ is approximately 0.1, and the mobile ion $M^+ = Na^+, K^+, Ag^+$) as model systems. In β-aluminas, ion conduction occurs over two non-equivalent lattice sites (Fig. 1a), and correlations persist over timescales corresponding to several (at least two) consecutive hops[7]. The vibrational modes of the mobile ions fall between 0.7 and 3.0 THz, depending on the mobile ion[36,37]. To match the direction of ionic hopping, the pump electric field $E$ is perpendicular to the crystalline $c$ axis and parallel to the two-dimensional conduction planes. Figure 2a shows the time traces of the terahertz-pumped transient birefringence in β-aluminas at 300 K. All samples show both oscillatory and non-oscillatory responses at 300 and 620 K (Extended Data Fig. 1). Such incoherent, non-oscillatory relaxation has been previously observed in liquids[27,29,38,39] and solids[40], but was attributed to overdamped rotations or librations, which are absent in β-aluminas.

Whereas the magnitude of the birefringence response scales with the square of the pump field (Supplementary Fig. 3), as expected for a Kerr effect signal, we do not observe any spectral changes with increasing pump fluence in either TKE or terahertz transmission experiments (Supplementary Figs. 3 and 5), indicating that we are probing a response intrinsic to the material and not a threshold-dependent response only relevant at high fields[41]. The non-oscillatory relaxation is substantially slower than any vibrational components. Therefore, we also rule out nonlinear phonon coupling[42,43]. Otherwise, the vibrational components of the TKE signals (Fig. 2b) are consistent with established far-infrared and Raman modes[37] of mobile ions.

We investigate the possible connection between temperature-activated hopping conduction and the picosecond TKE relaxation by varying the temperature of the samples. In all materials, the signals from thinner (less than 30 μm) versus thicker (100–300 μm) samples match after the short-time oscillations dephase (Extended Data Fig. 2). The subsequent non-oscillatory relaxation represents a bulk material response. We analyse a thick sample of K$^+$ β-alumina (Fig. 3a) here to

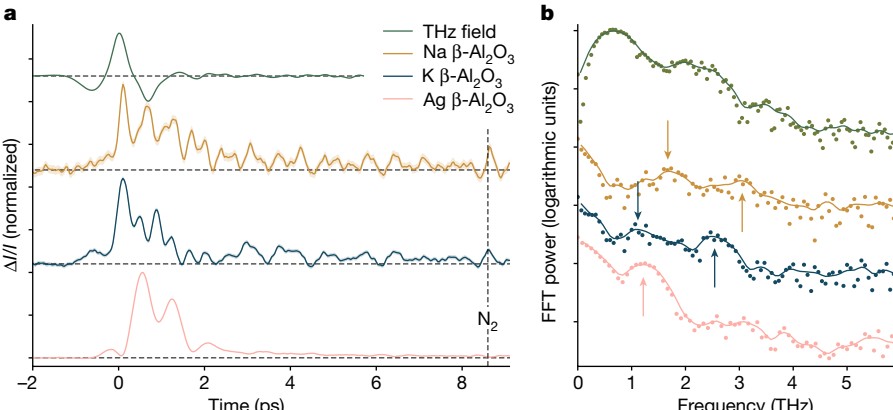

**Fig. 2 | TKE in β-alumina ion conductors. a**, Time-domain traces of transient birefringence in β-alumina ion conductors: Na⁺ (orange), K⁺ (blue), Ag⁺ (pink) and electro-optic sampling trace of the pump field (green), smoothed with a 30 fs Gaussian window. The shaded regions correspond to ±1 s.e. of the mean signal at each time delay. The labelled feature at 8.4 ps is the rotational coherence of atmospheric nitrogen[50] within a Rayleigh length (greater than 500 μm) of the samples (below 30 μm in thickness). **b**, Fast Fourier-transform (FFT) power spectra (points) of the signals in **a**, and smoothed with a Gaussian filter of 0.1 THz s.d. (lines), with peaks highlighted by arrows.

highlight this long time signal. We model the response as the sum of an instantaneous response arising from intrinsic nonlinearity and a mismatch in optical constants between the pump and probe frequencies[44,45] (Supplementary Notes 1 and 2 and Extended Data Figs. 3–5), and single-exponential decay, shown together as dashed lines. The slow non-oscillatory component is absent in the non-resonant optical Kerr response (Extended Data Fig. 6), and the residuals of the TKE fit (Extended Data Fig. 7a) show the same frequency, approximately 2 THz, as the non-resonant signal. Together, this suggests that both pumps (THz and optical) excite coherent ionic vibrations, but the terahertz pump excites an extra response that decays incoherently. For K⁺ β-alumina, this non-oscillatory relaxation accelerates from

approximately 10 ps at 300 K to 3–4 ps at 620 K (Fig. 3b), with an activation energy of 40 ± 5 meV. In ambient atmosphere, the apparent time constants are slightly slower due to the overlaid signal from atmospheric water vapour[27] (Supplementary Fig. 1), which is slower than the sample response and is not temperature activated.

Eliminating several other plausible assignments, we propose that the liquid-like picosecond-timescale component of the TKE response arises from the incoherent hopping of mobile ions. We use large-scale molecular dynamics simulations with a pulsed electrical field[27,46,47] mimicking the experimental THz pump in frequency and magnitude to verify this hypothesis. Following the simulated terahertz pump, the anisotropy of hopping rates is calculated from the times and crystallographic

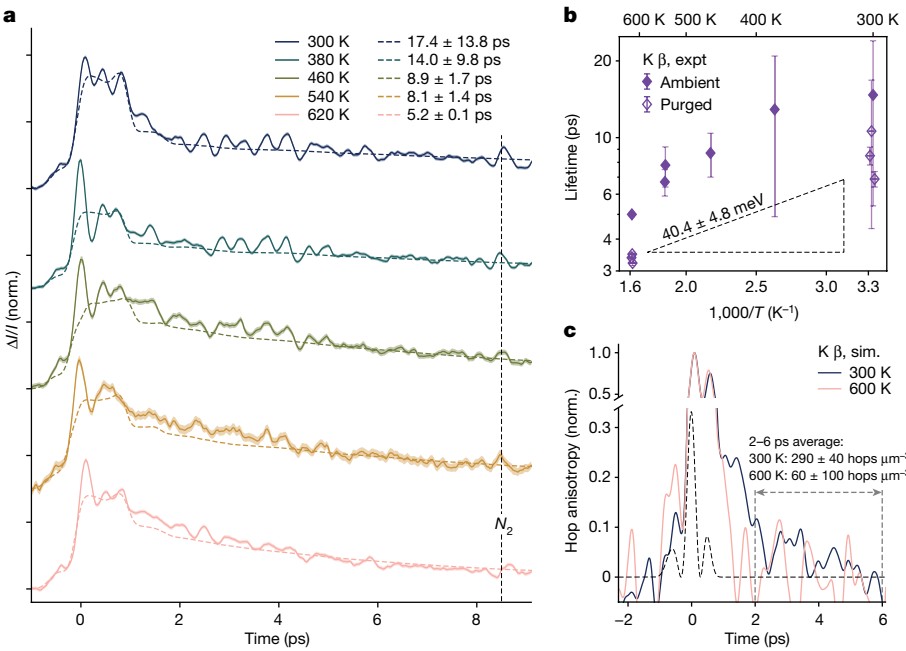

**Fig. 3 | Temperature dependence of the long-lived TKE in K⁺ β-alumina.**
**a**, Time-domain traces of transient birefringence in K⁺ β-alumina, measured in ambient atmosphere (solid lines), normalized and offset for clarity. The shaded regions correspond to ±1 s.e.m. Coloured dashed lines are fits to the sum of single-exponent relaxation and an instantaneous polarization. The labelled feature at 8.4 ps is the rotational coherence of atmospheric nitrogen. **b**, Time constants of single-exponential fits to the long-lived TKE component as

measured experimentally (expt, roughly 200 μm) in ambient atmosphere (filled symbols) and dry atmosphere (empty symbols). Error bars are ±1 s.e. of least-squares fitting. The dashed line at an activation energy of 40(5) meV is a fit to the purged-atmosphere measurements. **c**, Normalized anisotropy of hopping simulated (sim.) with molecular dynamics under applied electric field mimicking the experimental terahertz pump (Methods) at 300 K (blue) and 600 K (pink). The black dashed line shows the square of the simulated electric field.

directions of all hops (Fig. 3c). The pump selectively accelerates mobile ions (Supplementary Note 4) and drives hopping along its electric field. In K⁺ β-alumina at 300 K, the anisotropy of hopping is distinguishable from zero until more than or equal to 6 ps after the peak applied field (Fig. 3c, blue), but relaxes to zero by 3–4 ps at 600 K, in agreement with experiment. The simulated relaxation of the hopping anisotropy is only slightly faster than the experimental TKE relaxation. We conclude that the liquid-like picosecond-timescale TKE response in β-aluminas is indeed a signature of anisotropy in the hopping of mobile ions caused by the pump, the decay of which is heat-activated with an energy of $40 \pm 5$ meV in K⁺ β-alumina (Fig. 3b).

## Correlations and memory in ion hopping

We now discuss the physical meaning of the TKE signal. The contribution of hopping to the TKE signal for any time delay scales with the hopping rate $H$, weighed by the hop directions relative to the pump's electric field. Hopping rates both parallel $(+v_\parallel)$ and antiparallel $(-v_\parallel)$ to the pump contribute positively to the TKE signal, whereas hopping orthogonal to the pump contributes negatively as $\pm v_\perp$ (Fig. 1c). The existence of the slow TKE signal signifies a mismatch between $v_\parallel$ and $v_\perp$, and implies that directions of consecutive hops are correlated. This is consistent with previous simulations[6,7]: following pump-driven hops at $t_0$ (Fig. 1d), the mobile ions have a preference to hop backwards (correlation factors $f, f_1 \ll 1$ using solid-state ionics nomenclature[14,15]), which randomizes only slowly. TKE measures the decay of this preference and verifies that the atomistic mechanism of ion transport in β-aluminas differs from a random walk.

The hopping component of the TKE signal is temperature activated, but the activation is much smaller than for low-frequency conductivity ($40 \pm 5$ meV in Fig. 3b versus more than 200 meV, ref. 48). However, any hop in isolation is not timescale-dependent. The difference between hopping at picosecond versus macroscopic timescales is not within individual hops, but in the mutual information between successive hops (Supplementary Note 5), conceptually similar to a viscoelastic effect. Small activation energies measured with TKE are consistent with the slow loss of such correlation via production of entropy (Fig. 1b), being the origin of increasing measured activation energies toward macroscopic timescales[22]. At picosecond timescales, as probed here, even though hopping is triggered, correlations persist throughout the measurement, and only partial activation is measurable. Conversely, the full activation for macroscopic conduction is measured only at timescales sufficient for the decay of all correlations over multiple hops.

## Random-walk ion conduction in K⁺ β″-alumina

All β-aluminas show the picosecond 'tail' in the TKE response, implying that hopping remains correlated for longer than the TKE response is measurable[7]. This is expected because of the presence of non-equivalent lattice sites (Fig. 1). To verify the above interpretation, we next seek a control system in which ion transport could proceed via a true atomistic random walk. In such a system, any TKE tail could be expected to vanish rapidly. In the closely analogous β″-aluminas ($M_{1.67}Al_{10.67}Li_{0.33}O_{17}$, where the mobile ion $M^+ = Na^+, K^+, Ag^+$) all lattice sites for ion hopping are equivalent (Fig. 4a, inset). The β″-aluminas are expected to have true random-walk conduction at elevated temperature, but not at 300 K (ref. 7). In the TKE response of K⁺ β″-alumina, we observe a long-lived non-oscillatory relaxation at 300 K (Fig. 4a), as in the β-aluminas. At 620 K, this component is absent, which suggests the absence of long time correlations, and a more field-following response is observed overall.

In agreement with experiment, the simulated anisotropy of hopping following a 0.7 THz pump shows a long-lived relaxation at 300 K, but not at 600 K (Fig. 4b). Simulation reproduces the most important feature

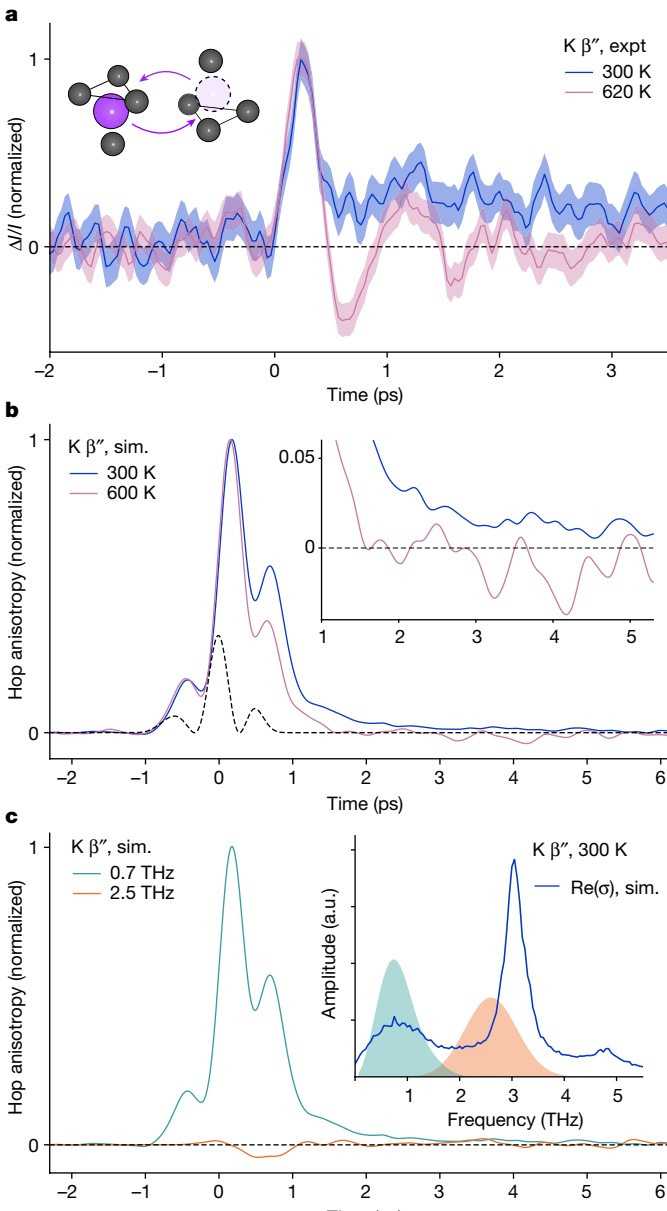

**Fig. 4 | TKE in K⁺ β″-alumina. a**, Normalized time-domain transient birefringence in thin polycrystalline K⁺ β″-alumina at 300 K (blue) and 620 K (pink). The shaded regions are ±1 s.e.m. The inset shows that in β″-aluminas, mobile ions (purple) hop (arrows) between equivalent four-coordinate sites. **b**, Normalized simulated (sim.) anisotropy of hopping directions in polycrystalline K⁺ β″-alumina under applied electric field centred at 0.7 THz mimicking the experimental pump, 300 K (blue) and 600 K (pink). The black dashed line indicates the square of the simulated pump electric field. The inset zooms in on the post-pump times and simulated hopping anisotropy at 300 K is non-zero for more than or equal to 5 ps. **c**, Simulated anisotropy of hopping in polycrystalline K⁺ β″-alumina at 300 K under applied electric fields centred at 0.7 THz (teal) and 2.5 THz (orange). The inset shows simulated in-plane optical K⁺ conductivity of K⁺ β″-alumina in the terahertz region (blue). Peaks at 1, 3, and 5 THz agree with the literature[49]. The spectra of simulated pumps with centre frequencies 0.7 THz (teal) mimicking the experimental pump (**a**,**b**), and 2.5 THz (orange) are shaded.

of experimental TKE: a picosecond-timescale anisotropy of hopping at 300 K that disappears by 600 K. Both experiment and simulation are consistent with a change of transport mechanism: from correlated at 300 K to a true random walk by 600 K (ref. 7). We conclude that the non-oscillatory picosecond-timescale relaxation of the TKE indeed

probes the decay of orientational memory in ionic hopping at the level of individual ionic hops.

## Attempt frequencies for ion hopping

Finally, we use the TKE measurements and simulations of $K^+$ $\beta''$-alumina to identify the vibrational origin of ionic hops: the attempt frequency $v_0$. To cause an anisotropy in hopping, a pump field must excite the vibrational modes $Q_i$ that couple directly to hopping and trigger hopping aligned with the applied field. This must happen before thermalization, or else pump-driven heating enhances hopping rates in all directions isotropically. Having established the correspondence of the experimental TKE and simulated anisotropy of ionic hopping, we use further molecular dynamics simulations as in silico TKE at pump frequencies presently inaccessible to us experimentally. We simulate the anisotropy of hopping in $K^+$ $\beta''$-alumina following a 2.5-THz pulse that overlaps with the known strong infrared-active vibration of the $K^+$ ions in the conduction plane at 3.0 THz (ref. 49) (Fig. 4c, orange). Despite the material absorbing nearly the same amount of energy from simulated 0.7 and 2.5 THz pulses (Supplementary Fig. 6), and despite the 3.0 THz vibration being coherently driven by the 2.5 THz pulse (Supplementary Fig. 7), the 0.7 THz pulse creates an approximately 20-times stronger anisotropy of hopping than the 2.5 THz pulse (Fig. 4c). This suggests that the pulse centred close to 1 THz couples directly to hops, whereas the 2.5 THz pulse heats the material isotropically. We conclude that the attempt frequency in $K^+$ $\beta''$-alumina is roughly 1 THz. The excitation of this vibration is evident in the experimental TKE response (Fig. 4a and Extended Data Fig. 7b).

The attempt frequencies in $\beta$-aluminas can be similarly identified once the distinct dynamics of bound defect clusters and non-equivalent lattice sites are disaggregated (Supplementary Note 6). In Na $\beta$-alumina, the simulated attempt frequencies are $1.4 \pm 0.2$ THz for the dominant ion-pair hopping mechanism and approximately 2.3 THz for the more rare single-ion hopping. The latter corresponds to the known vibration[37] commonly taken as the attempt frequency[41] and is verified here with THz transmission (2.1 THz, Supplementary Fig. 5 and Supplementary Note 6). Instead, the lower frequency vibration, highlighted by neutron scattering at 5–6 meV (ref. 36), drives most hopping at 300 K, despite constituting a minor part of the vibrational density of states for the mobile ion.

## Conclusions

In summary, we have used impulsive near-resonant terahertz excitation to trigger ionic hopping in solid-state ionic conductors. Picosecond transient birefringence arises from correlations in the hopping of mobile ions. Therefore, we establish TKE, a nonlinear optical measurement, as a direct probe of path dependence in ionic transport[7,8]. TKE highlights both the vibrational origination of ionic conduction, and the slow loss of memory during diffusion. Ionic conductivity and its activation reach their low-frequency limits only at timescales sufficiently long to scramble all memory within the system. The transport of ions can be characterized by a random walk only at timescales longer than the persistence of correlations, but such a phenomenological random walk may not correspond to a true atomistic one. In other words, macroscopic measurements cannot be interpreted in terms of atomistic quantities without accounting for such correlations. By distinguishing rapid ion hops from the persistent correlations connecting them, our work demonstrates the correspondence between thermodynamics and information for thermally activated mass transport.

In addition to probing the mechanisms of ionic conduction, the correlation effects we highlight are of importance for transport under strong driving forces, at short timescales and in confined dimensions[9,10], such as in switching applications[3]. This study provides a framework for the use of nonlinear optical techniques to probe the atomistic mechanisms of non-equilibrium transport phenomena, which will be highly valuable in the development of energy technologies such as solid-state batteries, and in the related fields of nanofluidics, engineering phase transformations and neuromorphic computing.

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

# Methods

## Sample preparation

Single crystals of Na β-alumina were graciously shared with us by O. Kamishima. They were ion-exchanged to Ag and K in molten nitrates until no mass change was detectable, at least 3 days. For Ag, the process had to be repeated. For K, mixed nitrate compositions were used first to avoid mechanical damage from the thermodynamically favourable ion exchange. The *c* lattice constants for the Na, Ag and K β-alumina samples were 22.53, 22.49 and 22.73 Å, respectively. Polycrystalline K β″-alumina was purchased from Ionotec Ltd as a pellet. To produce thin samples (roughly 5–30 μm), single crystals and the polycrystalline material were hand-polished using a T-tool and then dried. Double-side polished sapphire (0001) was purchased from MTI Corp.

## TKE

The output of a Ti:sapphire oscillator (Coherent Micra) is amplified (Coherent Spitfire) to roughly 4.2 mJ at 1 kHz, and pulse width optimized to maximize the peak terahertz field at the sample, roughly 150 fs full-width at half-maximum. Of the output, 99% is used to generate terahertz pulses (roughly 6 μJ) via optical rectification in lithium niobate using the tilted pulse front method[32,51]. The terahertz pulse is focused on the sample using a pair of off-axis parabolic mirrors. Peak field amplitudes at the sample position were about 700 kV cm$^{-1}$ in ambient atmosphere, and about 600 kV cm$^{-1}$ in purged (at or below 0.1% RH) atmosphere. For the probe pulse, 1% of the amplifier output is used and polarized at 45° from the pump pulse, overlapped with the pump at the sample position and passed through a quarter-wave plate and a Wollaston prism. The birefringence of the transmitted probe is measured with two photodiodes in a balanced detection scheme. The terahertz waveform at the sample position is measured with a free-standing uniaxially poled mixture of an electro-optic dye and amorphous polycarbonate polymer[52,53], and the peak field strength measured by electro-optic sampling in GaP(110). The time delay between pump and probe pulses is varied with a mechanical delay stage. For temperature control and measurements at elevated temperature, a transmission-mode heating stage (Linkam) was used without windows to eliminate their contributions to transient birefringence. The transient birefringence of ambient and purged air (Supplementary Note 2) was measured by aligning to a thin sample and removing the sample. Most experimental data are adapted from A.D.P.'s doctoral dissertation[54].

## Terahertz transmission

Thin (30 μm or below in thickness) samples are mounted on thin metallic pinholes of diameter 1 mm, fully covering the pinhole. The terahertz pulse transmitted through the sample is focused with a second pair of off-axis parabolic mirrors and sampled with the free-standing film of an electro-optic dye and amorphous polycarbonate polymer[52,53]. A time delay sweep of an empty pinhole is measured following every time delay sweep measuring the sample. Optical conductivity was fit to the time-domain spectra assuming a slab geometry[55,56] using a nonlinear fitting procedure initialized with $\tilde{n} = 2 + 0.5j$ as the complex refractive index. This yielded stable fitting without the single-pass assumption.

## Optical Kerr effect

The output of a Ti:sapphire oscillator (Coherent Micra) was amplified (Coherent RegA) at 100 kHz to about 1 μJ and compressed to about 50 fs full-width at half-maximum (the width of the coherent artefact in the optical Kerr effect). The pump and probe pulses were overlapped on the sample so that probe $E \perp c$ for single crystals, and pump roughly 15° off. The transmitted probe was detected using a pair of Si photodetectors in a balanced detection scheme. Frequencies of oscillatory components in the optical Kerr effect signals were fit using linear prediction fitting[57] optimizing for least-squares error over the start and end points, which serve as hyperparameters.

## Post-measurement characterization

The thin K β″-alumina sample was subjected to further high-intensity kHz-pulsed 800 nm illumination, including hitting the Inconel support. The heating of the Inconel support during this procedure created a small hole in the sample where it was thinnest. The local phase composition of the polycrystal K β″-alumina sample was subsequently analysed with X-ray microdiffraction (Bruker D8 Venture). All samples were imaged with an optical microscope to check for damage.

## Steady-state molecular dynamics

Steady-state classical miolecular dynamics simulations were carried out in LAMMPS[58] using Buckingham pairwise potentials with Coulombic interactions as described previously[7]. The vibrational density of states was calculated from the velocity autocorrelation functions for the mobile ions, and the optical conductivity from the velocity autocorrelation function for the centre of mass of the mobile ions[59].

## Molecular dynamics with terahertz pumping

An impulsive electrical field was included in the molecular dynamics simulations to mimic the experimental terahertz pulse[27,28,46,47]. Peak field $E = 300$ kV cm$^{-1}$ polarized parallel to the conduction plane was used to account for a front-surface reflection of roughly 50%. Charge-compensating defects were placed in the scaled-up simulations using the same procedure as described previously[7]. To sample hopping events, 800 randomized iterations per temperature point, with 4,896 mobile ions each, were used for β-aluminas. For K β″-alumina, the simulations contained 5,940 mobile ions each. Each iteration started with an anneal to 1,000 K to randomize starting positions of the mobile ions, and a short equilibration at the simulation temperature in the constant-volume (NVT) ensemble. The electric field was applied in the microcanonical (NVE) ensemble and the simulations propagated for 15 ps, enough for mobile ions and the host lattice to come to the same temperature. For K β″-alumina, polycrystal orientations to match the experimentally available sample were averaged by changing the polarization of the simulated field over 12 angles in increments of 30°. No notable dependence on the pump polarization within the conduction plane was observed in simulations of β-aluminas. The hopping statistics were extracted from all simulations and analysed as described previously[7]. The direction of each hopping event was taken as the vector connecting the start and end crystallographic sites of the hop. The anisotropy of hopping was calculated by weighing the hopping directions by $(\cos\theta)^2 - \langle(\cos\theta)^2\rangle$, where $\theta$ is the angle between the simulated pump field and the hop direction and $\langle(\cos\theta)^2\rangle = 0.5$ in two dimensions.

## Data availability

Experimental data, example computational data, and analysis scripts are available at https://doi.org/10.5281/zenodo.8169681 (ref. 60).

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

**Acknowledgements** This work was supported by the US Department of Energy, Office of Basic Energy Sciences, Division of Materials Sciences and Engineering (contract no. DE-AC02-76SF00515). M.S.I. and J.A.D. gratefully acknowledge the EPSRC (Engineering and Physical Sciences Research Council) Programme Grant 'Enabling next generation lithium batteries' (no. EP/M009521/1). J.A.D. gratefully acknowledges the EPSRC (grant no. EP/V013130/1), Research England (Newcastle University Centre for Energy QR Strategic Priorities Fund) and Newcastle University (Newcastle Academic Track Fellowship) for funding. We are indebted to O. Kamishima (Setsunan University) for sharing single-crystalline samples of Na β-alumina with us.

**Author contributions** The probing of ionic transport with nonlinear techniques was proposed by A.M.L., M.T., S.W.T. and A.D.P. Simulations incorporating the terahertz pump were proposed by A.M.L., J.A.D. and M.S.I. A.D.P. and M.C.H. performed the TKE and terahertz transmission measurements. A.D.P., S.W.T. and M.T. performed the OKE measurements. A.D.P. analysed the experimental data. A.D.P. performed and analysed the molecular dynamics simulations with advice from J.A.D. and M.S.I. A.M.L. advised and supervised the work. All authors contributed to the writing of the manuscript.

**Competing interests** The authors declare no competing interests.

## Additional information
**Correspondence and requests for materials** should be addressed to Andrey D. Poletayev or Aaron M. Lindenberg.

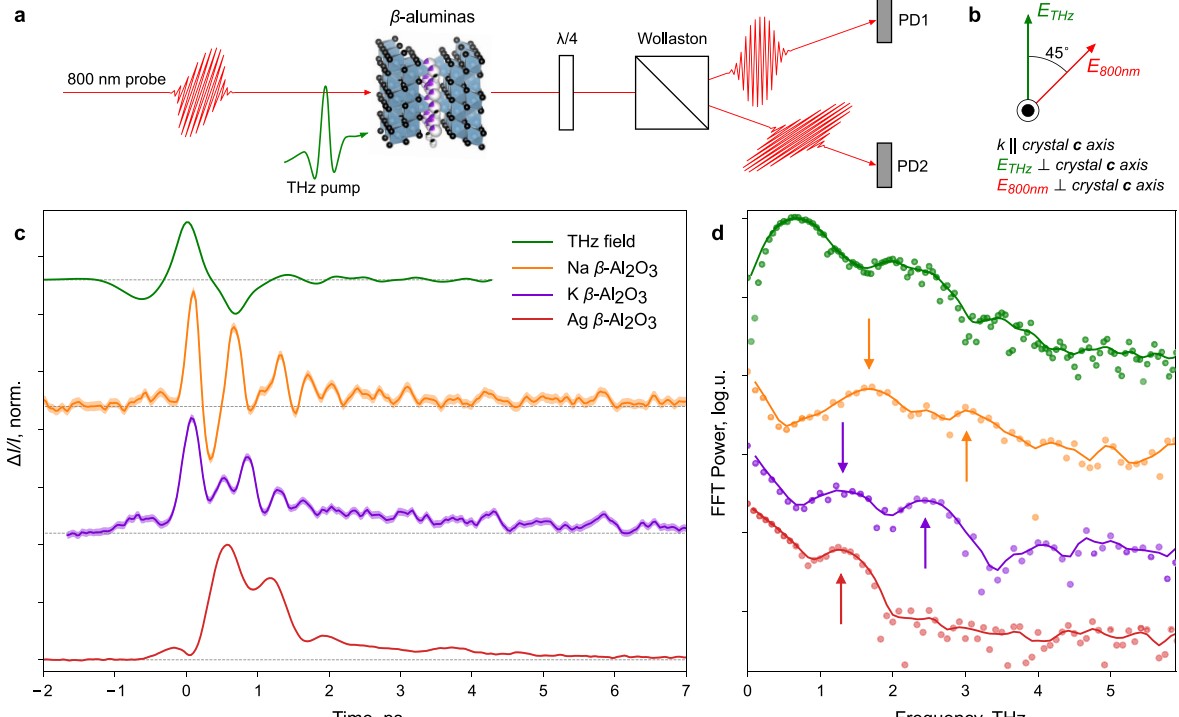

**Extended Data Fig. 1 | Terahertz Kerr effect (TKE) in $\beta$-aluminas at elevated temperature. (a)** Schematic of the transient birefringence experiment: terahertz pump (green) perturbs mobile ions (purple) in the ionic conductor. The 800 nm probe pulse (pink) delayed by a time $\Delta t$ is split with a quarter wave plate and a Wollaston prism. The polarization rotation of the probe pulse is detected with a pair of Si photodiodes in a balanced detection scheme. **(b)** Polarizations of the pump and probe pulses relative to each other and the $\beta$-alumina crystals. **(c)** Time-domain traces of transient birefringence in thin single crystals of $\beta$-aluminas at 620 K: $Na^+$ (orange), $K^+$ (purple), $Ag^+$ (dark red), and electro-optic sampling trace of the pump field (green). The shaded regions correspond to ±1 s.e. of the mean signal at each time delay. **(d)** Fourier transform power spectra (points) of the signals in **(c)** and smoothed with a Gaussian filter of 0.1 THz st. dev. (lines), with peaks highlighted by arrows.

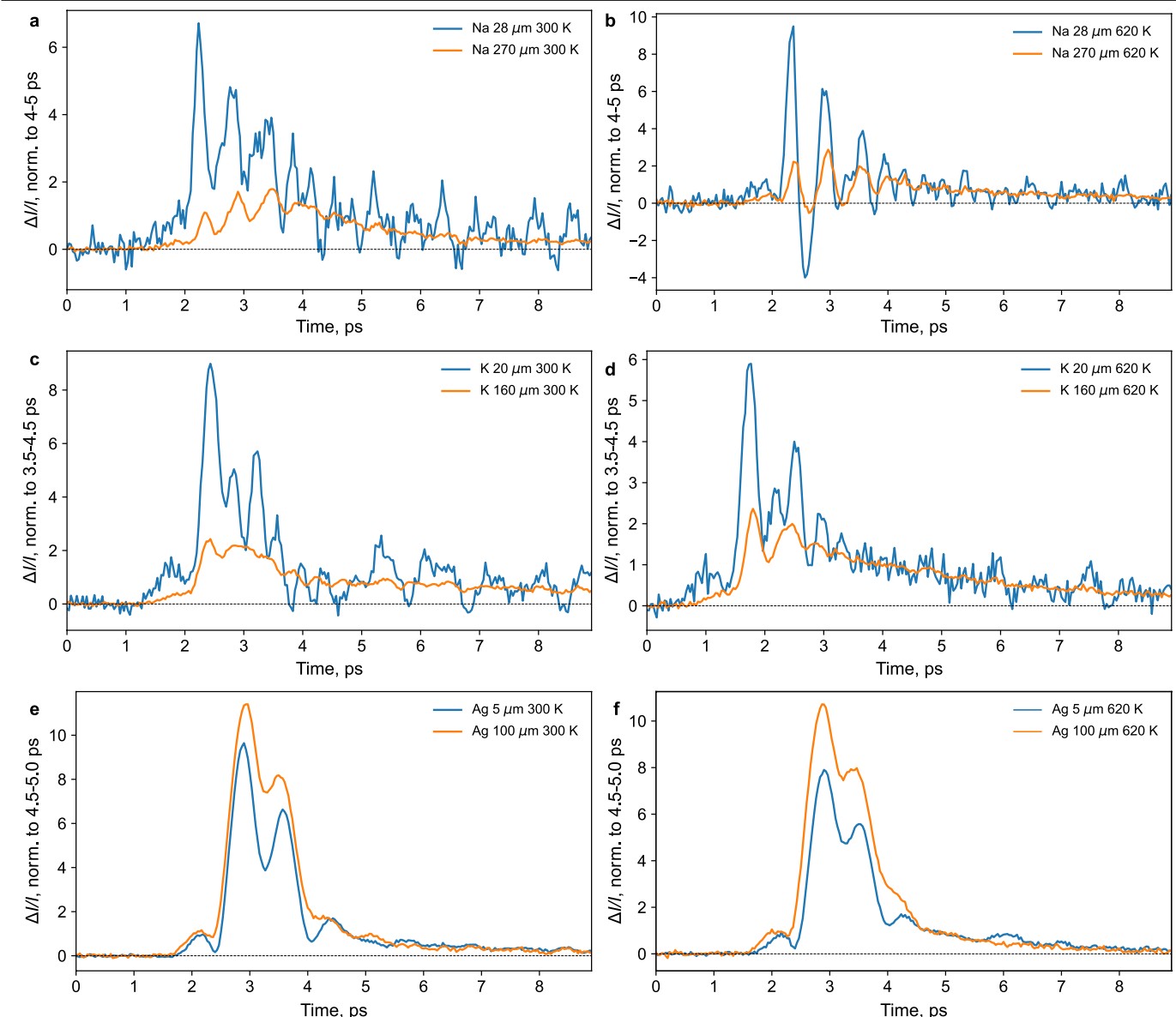

**Extended Data Fig. 2 | TKE in β-aluminas with varying sample thickness.**
TKE signals for thick (≥100 μm) and thin (≤30 μm) crystals of Na (**ab**), K (**cd**), and Ag (**ef**) β-alumina, normalized to their values at a pump–probe delay time when short-time oscillatory signal components have decayed. For all materials at 300 K (**ace**) and 620 K (**bdf**) the longer-time relaxation of the signals overlap: their kinetics are independent of sample thickness. Since the TKE signal is phase accumulated over the sample thickness, thicker samples yield lower-noise signals at long time delays.

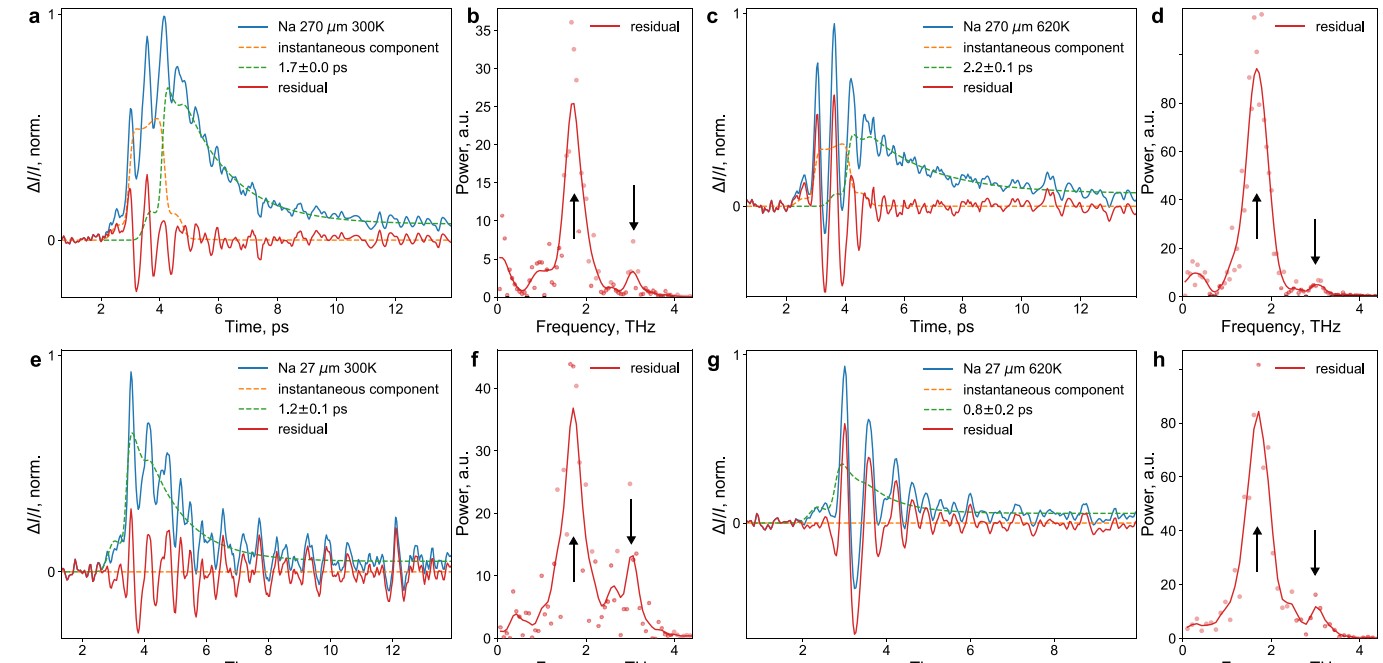

**Extended Data Fig. 3 | Fitting TKE signals of Na $\beta$-alumina.** Each signal (blue in **aceg**) was measured in a purged atmosphere and modelled as the sum of an instantaneous component (orange) and a single-exponential relaxation (green) with a long-time constant (Supplementary Notes 1 and 2). For the thin sample (**e-h**), the strength of the instantaneous component was small. The strong vibrational component in the thick-sample signal (a-d) precluded unambiguous fitting at short times, so the exponential component was fit to the long-time part of the signal. The residuals (dark red) are oscillatory. Their Fourier transforms (**bdfh**), plotted with 0.1-THz Gaussian-filter smoothing, show main frequency components at 1.8 and 3.0 THz in agreement with non-resonant OKE (3.0 THz, Extended Data Fig. 6), literature, and simulation (1.7 THz, Fig. S7).

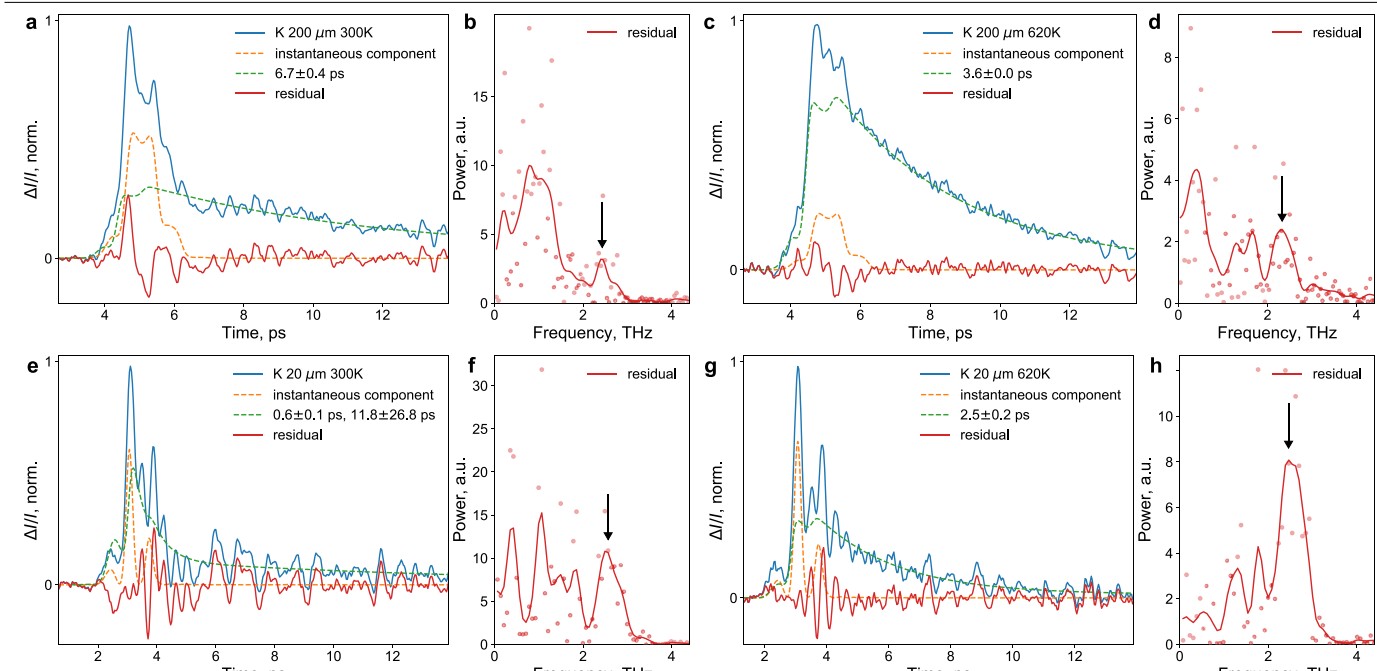

**Extended Data Fig. 4 | Fitting TKE signals of K $\beta$-alumina.** Each signal (blue in **aceg**) was measured in a purged atmosphere and modeled as the sum of an instantaneous component (orange) and a single-exponential relaxation (green) with a long-time constant (Supplementary Notes 1 and 2). For the thin sample (**e-h**), the instantaneous component was non-negligible, and at 300 K a weakly determined second exponential was identifiable. The residuals (dark red) are oscillatory and most interpretable in the signals from the thin sample (**e-h**). Their Fourier transforms (**bdfh**), plotted with 0.1-THz Gaussian-filter smoothing, show main frequency components at ≈1.2, ≈2.0, and ≈2.7 THz in agreement with non-resonant OKE (2.0 THz, Extended Data Fig. 6), literature, and simulation (1.3 THz and 1.9 THz, Fig. S7).

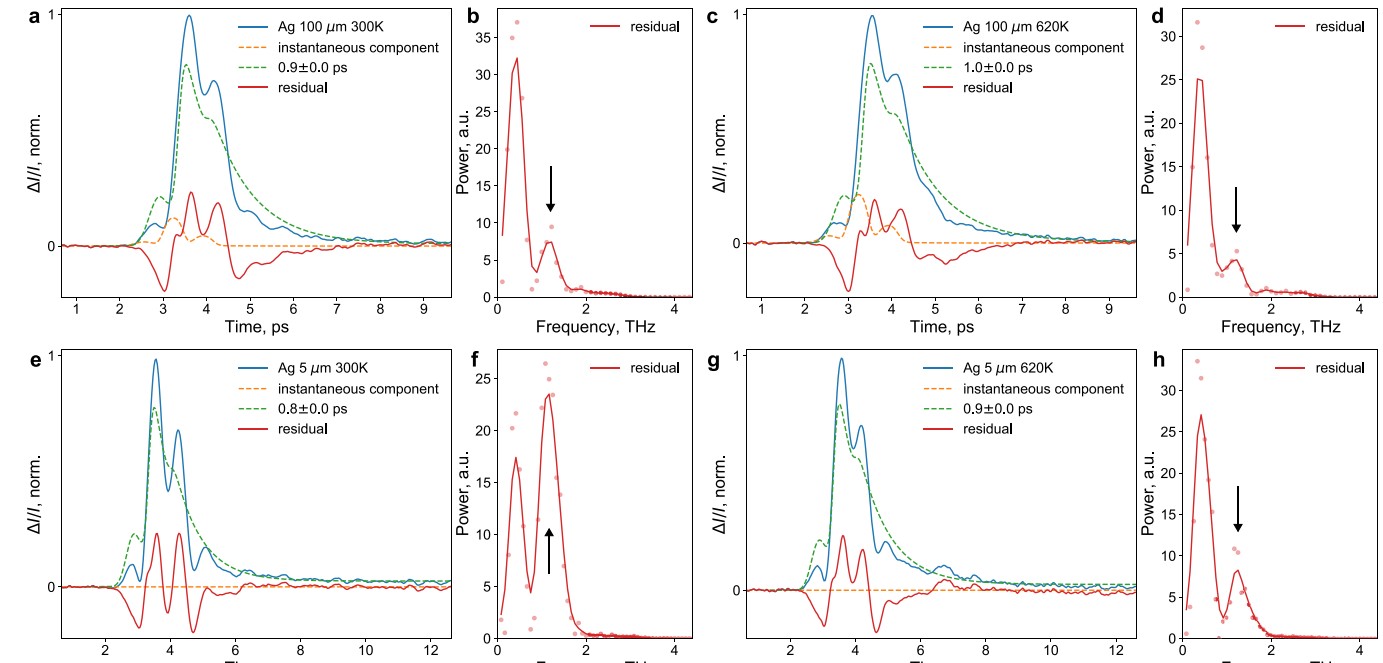

**Extended Data Fig. 5 | Fitting TKE signals of Ag β-alumina.** Each signal (blue in **aceg**) was measured in a purged atmosphere and modeled as the sum of an instantaneous component (orange) and a single-exponential relaxation (green) (Supplementary Notes 1 and 2). For the thin sample (**e-h**), the instantaneous component was negligible. The Fourier transforms (**bdfh**) of residuals (dark red), plotted with 0.1-THz Gaussian-filter smoothing, show a frequency component at ≈1.2 THz in agreement with literature (infrared-active mode at 1.0-1.2 THz) and simulation (1.0 THz, Fig. S7). The signal is very similar between the thick and thin samples (Extended Data Fig. 2), consistent with it arising from a thin layer of the material and consistent with strong absorption (Supplementary Note 3).

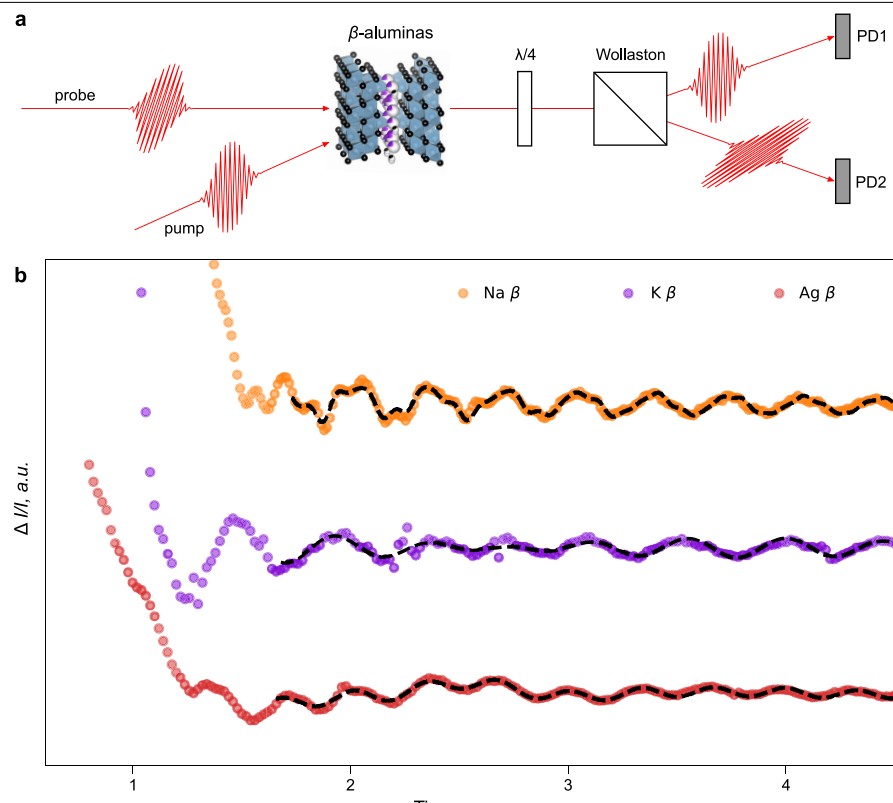

**Extended Data Fig. 6 | Non-resonant optical Kerr effect in $\beta$-aluminas.** Schematic of the experimental configuration (**a**) and signals from thick Na (orange), K (purple), and Ag (dark red) $\beta$-alumina crystals (**b**) at positive time delays following a strong coherent artefact at zero time delay. The frequency components are extracted from linear prediction fitting results (Methods) shown as black dashed lines.

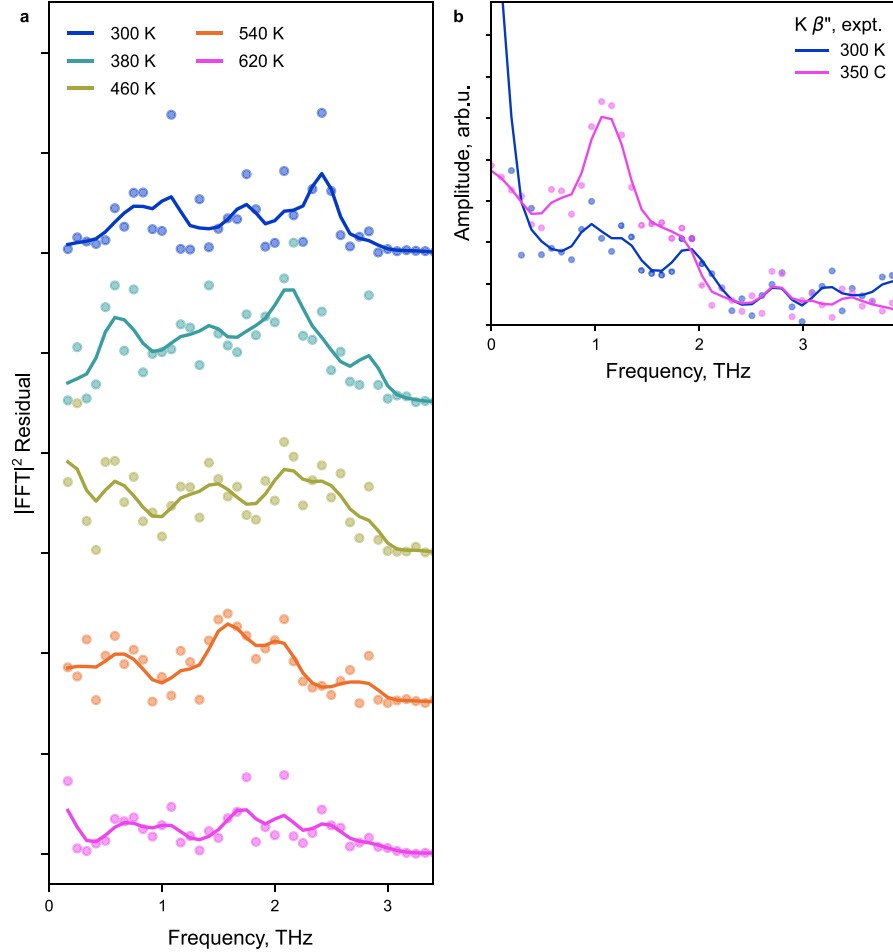

**Extended Data Fig. 7 | Fourier Transforms of TKE Signals in K⁺ β- and β″-alumina.** (**a**) Fourier transforms of oscillatory residuals in TKE signals of thick crystals of K β-alumina measured in ambient atmosphere (Fig. 3a). (**b**) Fourier transforms of the TKE signals of thin polycrystalline K⁺ β″-alumina (Fig. 4a).