## [Peer Review File · Nature]

Manuscript Title: The Persistence of Memory in Ionic Conduction Probed by Nonlinear Optics

Reviewer Comments & Author Rebuttals

Reviewer Reports on the Initial Version:

Referees' comments:

Referee #1 (Remarks to the Author):

A. Summary of the key results

The authors report on a combined experimental and theoretical study of the materials response of ionic conductors $A^+ \beta$ -Alumina with $A = Ag, Na, K$ and $K^+ \beta''$ -Alumina upon excitation with a Terahertz (THz) pulse. Experimentally, the transient birefringence response is probed optically on the picosecond timescale for different materials, sample thickness, temperature, atmospheres, pump field as well as using an optical pump pulse. This allows the identification of (long-lived) anisotropic ion relaxation contributions to the probed Terahertz-Kerr-effect signal and its activation energy (in $K^+ \beta$ -Alumina exclusively). This ion hopping anisotropy expressing a deviation from a true random walk mechanism was then also captured in molecular dynamics with simulated Terahertz pumping. The frequency of this simulated pump pulse was additionally altered to capture the attempt frequencies of ionic hops.

B. Originality and Significance

The application of the nonlinear optics method probing ion diffusion properties by THz irradiation gives novel and complementary insight into ion jump dynamics in the time domain otherwise experimentally inaccessible. This adds to a fundamental insight into solid state ion transport an increasingly discussed field in multiple energy storage applications.

C. Data & Methodology:

Figure 4 or $K^+ \beta''$ -Alumina measurements. Here the picosecond range shown is much shorter than the ones shown before. While lower signal to noise ratio is present, longer times would be beneficial to show whether the TKE signal recorded at 300K does fully decay. Suggestion equal time scale to simulation in Figure 4b.

D. Appropriate use of Statistics and treatment of uncertainties

Different values for simulated attempt frequencies for $Na \beta$ -Alumina; 1.3 THz in Supporting Information and 1.2 THz for ion pair hopping in the main text. Is an interval or generally an estimation of uncertainties for the frequencies given a more reasonable approach to report those values?

Consistency about the uncertainty of activation energy of the decay process in $K^+ \beta$ -Alumina (Caption Figure 3 and "Picosecond hopping dynamics in β -aluminas" / "Correlations and Memory in Ion Hopping" section); A triplicate measurement in purged atmosphere leads to uncertainties, which could be propagated to a linear fit giving rise to uncertainties of the "guide to the eye" 45 meV line. You give " ≈ 45 meV" and " $\approx 40 - 50$ meV". At least give reason/or state estimated " ± 5 meV" uncertainty.

E. Conclusions: robustness, validity, reliability

Equation 2; (Polarization = Dipoles+ Int(hopping rates)); You correctly describe a non-zero

contribution of the integral for hopping in a preferential direction. You describe this to be the case in your experimental setup until the pump drives those. If equilibration at elevated temperatures or any other than a radial temperature gradient is applied an anisotropy of hopping rates might also be present due to temperature, which is most likely negligible but maybe noteworthy.

Can the observed anisotropy not only be interpreted as the lifetime of an occupied interstitial A site (excited state) comparable to transition state theory? A preferred back hopping into the B site (and not the B' site) might additionally be indicating that not all neighboring B' sites are unoccupied. This would then be a "common" statistical site blocking effect, while "single hops" are probed still an average over a large amount of "single hops" is probed. The lower activation energy of the decay is then only the barrier out of the interstitial A site, and not the B → A site, which should be rate determining (largest activation barrier). This may explain why the Ag⁺ and Na⁺ β-Aluminas do not show a clear activation behavior (as far as I can tell) from the 300 K vs. 620 K measurements (Extended Data Figures 3 and 5) as also stated in Supplementary note 4 more easily. And in the slower ionic conductor K⁺ β-Alumina may however be an activated process for a bit larger energetic barriers for A → B sites (as indicated in longer lifetimes).

Maybe decipher A→B and B→A site hops from simulation.

F. Suggested Improvements: experiments, data for possible revision

Equations should be numbered (and/or fully included into the text.)

Extended Data Figure 2c; Is the normalization regime correct?

Extended Data Figure 3a; Temperature is given as 620K, maybe 300K?

Extended Data Figure 5e; Temperature is given as 3000K, maybe 300K?

Figure 3 or temperature dependent measurements. Those do not overlap with the ones depicted in extended data Figure 2 and 4. It would be beneficial, to give the atmosphere used in the caption of every Figure containing TKE signals.

G. References

Update Reference 7 (Supplementary Reference 10)

It may be helpful to add this reference (End of Section "Probing correlated Hopping" or Supplementary Note 6) of a theoretical study excitation of certain vibrational modes relevant for ion diffusion, as well as a differing (LAMMPS) temperature of host lattice and mobile ions (Supplementary Note 4): Gordiz et al. Enhancement of ion diffusion by targeted phonon excitation, Cell Rep. Phys. Sci. 2021, 2, 100431; DOI: 10.1016/j.xcrp.2021.100431

H. Clarity and Context

Considering Figure 1b and the discussion of a random walk: The concept of a random walk is only ever defined for a sequence of ionic jumps. Those can include correlation effects, which distinguish a random walk from a "true random walk" and are associated to the correlation factors (or the heaven ratio in solid state ionics) [H. Mehrer, Diffusion in solids, 2006]. I would encourage more carefully phrasing of paragraph 2 in the "models of Ion Transport" Section.

Referee #2 (Remarks to the Author):

Review of Poletayev et al "The Persistence of Memory in Solid-State Ionic Conduction Probed by Nonlinear Optics"

The authors report on a study of ion conduction in Na, K, Ag beta-aluminas using THz pump - optical probe experiments and atomistic modeling. In particular, the nonlinear optics measurements support that single-cycle terahertz pumps could trigger the hopping process of the mobile ions in several compositions of beta-aluminas, with the dynamic response of the sample probed via a transient birefringence signal or Terahertz pump Kerr Effect ('TKE').

Based on THz pump -800nm probe TKE measurements the authors investigate the anisotropy of the mobile (alkali metal) ions in the beta aluminas within mobile planes in the crystal, and seek to quantify the temporal decay of the response ('memory'), as well as quantify the attempt frequency parameters, an often used descriptor in fast ion diffusion, such as solid electrolytes. Further, the authors extend the discussion to connect the microscopic hopping processes with entropy production and theory of information.

Overall, I find the manuscript and the conclusions rather technical. In addition, prior results in the literature (cited by the authors) have already established that THz pulses can trigger ionic hopping processes in solid battery electrolytes such as beta-aluminas (especially [41] Minami et al. "Macroscopic Ionic Flow in a Superionic Conductor Na+ β -Alumina Driven by Single-Cycle Terahertz Pulses", Phys. Rev. Lett. 2020), so this aspect of the results is not in itself a key scientific advance. The beta aluminas have been extensively studied in past decades so the choice of materials does not bring much novelty.

Finally, a broader question arises concerning the relevance of the methods/results to solid electrolytes for battery applications. The material response detected in these ultrafast nonlinear optical experiments is under intense ultrafast THz pulses that bring the samples very far from equilibrium. It is fairly unclear whether the state of the material and its transient response to such excitation is truly relevant to battery applications, a connection the authors attempt to make.

In view of the highly specialized nature of the study, some lack of novelty in the key result and materials studied, and the somewhat stretched relevance to the behavior of solid electrolytes for practical batteries, I do not recommend the manuscript for publication in Nature. I believe the results would be better suited for a more specialized journal.

More technical comments:

- While some signal is apparent in the ultrafast THz pump-probe measurements, the details of response seem challenging to interpret reliably. For instance, it is not discussed how much the signal detected depends on details of the THz pulse shape. I also noted that some of the time traces exhibited time structure on fast scale (e.g. sharper/shorter features) than the pump itself, which was not explained (Fig 2, Extended data Fig 1).

- The separation of fast/slow response and the fitting procedure illustrated in extended data Fig 3,4,5 is not fully justified/detailed and brings uncertainty to the analysis. Some data look rather noisy (also extended data Fig 7).

- The MD simulations are performed using simple potentials re-used from the literature. It is not clear that these would be appropriate for the intensely excited sample. For example, the electric field could modify the height of the potential barrier, changing the potential and ion dynamics.

- The section reporting sample preparation in supplement is very brief. I found the degree of characterization of samples reported insufficient to establish the full ion exchange (in case of Ag, K), or the crystal quality (before and after intense THz pump experiments).

- In a number of places, the references cited are difficult to reconcile with the statements made. I would encourage the authors to be more specific in their use of citations and comparison with prior literature. Also, in several places "agreement" with prior reported values (such as ion frequencies) is stated, but the reader is not provided with all the values to directly assess the level of "agreement".

Referee #3 (Remarks to the Author):

I imagine that it might depend on the reader whether this paper is valuable or not. The significance of this work is also detailed in Sections 2 and 5.5 of the cited review paper (Ref 11). In general, ionic conduction can be divided into two factors; enthalpy migration and entropy migration. Both factors have been investigated in superionic crystals including β -alumina from the viewpoint of physics for a long time, but only the enthalpy migration has been investigated with conventional methods by many researchers and developers of solid state ionics. Recently, the contribution of entropy migration has attracted attention again for improving solid electrolyte in novel fuel cells and all-solid-state batteries. THz spectroscopy and ultrafast time-resolved measurements were shown in Ref 11 as recent probing tools. THz Kerr effect measurement, widely used to investigate polar liquids, is based on both combinations. In this way, the authors succeeded in observing a phenomenon peculiar to entropy migration. The experimental results are also very reliable. In particular, the authors carefully checked the group velocity mismatch between the pump and probe pulse. Unfortunately, the authors focused on the traditional superionic conductor of β -alumina, which might be considered by some readers as an increment of old experiments. I imagine that single crystallization is necessary in their demonstration to use near infrared probe pulses, which implies that their proposed technique might not be applicable to popular sintered solid electrolytes. The authors may suppress the appeal in this manuscript. However, I highly appreciate that their work is the first demonstration to evaluate ion dynamics in ionic conductors clearly on the picosecond time scale. Basically, I consider that this work deserves publication in Nature.

However, I consider that the comparison between picosecond ionic motion and long-time motion is too rough. The authors have to measure AC impedance for their samples and have to compare it carefully. This is also important for showing their sample information. We see the ionic conductivity

of Na β -alumina in many review papers, which shows the activation energy changing at 200°C. However, the authors estimated the activation energy from THz nonlinearity in a wide temperature range.

In addition, the authors discuss the THz induced birefringence caused by ions hopping to adjacent sites, but I suspect whether terahertz electric fields really cause ions to hop to adjacent sites. In electronic systems, the kinetic energy accelerated by the half-cycle of the AC electric field is an important parameter for nonlinear optics. I roughly estimated that the kinetic energy of ion driven by an electric field of 300 kV/cm at 0.5 ps in the case of Ag (atomic mass of ~ 108) is $1/2 mv^2 = 2e^2 E^2 t^2 / m = 0.4$ meV, which is much smaller than the estimated energy of potential barrier (several tens of meV). Intuitive explanation is needed that the observed transient dipole originates from ion hopping.

Author Rebuttals to Initial Comments:

We thank the editor and the three referees for their detailed comments which have helped to improve the paper. We are also pleased to see positive comments including “...*novel and complementary insight into ion jump dynamics in the time domain otherwise experimentally inaccessible. This adds to a fundamental insight into solid state ion transport*” (Reviewer 1); “*I highly appreciate that their work is the first demonstration to evaluate ion dynamics in ionic conductors clearly on the picosecond time scale. Basically, I consider that this work deserves publication in Nature*” (Reviewer 3).

In the response below, we show the reviewers’ comments in **bold font** and both **highlight and italicize** the changes to the manuscript.

Reviewer 1 (a): **Update Reference 7 (Supplementary Reference 10)**

We have updated Ref. 7 to: “*Poletayev, A. D., Dawson, J. A., Islam, M. S. & Lindenberg, A. M. Defect-driven anomalous transport in fast-ion conducting solid electrolytes. Nat. Mater. 21, 1066–1073 (2022)*”. We refer to it extensively throughout this response using the shorthand “Ref. 7”.

Reviewer 1 (b): **Experimentally, the transient birefringence response is probed optically on the picosecond timescale for different materials, sample thickness, temperature, atmospheres, pump field as well as using an optical pump pulse. This allows the identification of (long-lived) anisotropic ion relaxation contributions to the probed Terahertz-Kerr-effect signal and its activation energy (in K⁺- β -Alumina exclusively).**

We thank the Reviewer for their detailed reading of our manuscript. We would like to offer a few clarifying comments here regarding the high-frequency activation of hopping and TKE in Na⁺ and Ag⁺ β -aluminas and the universality of our observations.

The Reviewer correctly points out that the activation of the TKE signal is only readily apparent in our measurements of K⁺ β -alumina. Nonetheless, we expect a similar activation to be a feature of transport in all materials whenever the transport process is not represented by a random walk. Overall, the differences between the TKE responses of the three β -aluminas are in degree, not in kind. We return to this below.

Due to two factors, (1) faster hopping in those two materials, relative to the K⁺ material, leading to shorter overall TKE relaxation, and (2) smaller high-frequency activation energies leading to the need for stronger signal or longer relaxation times, such activation in Na⁺ and Ag⁺ β -aluminas is beyond our current capacity to unambiguously distinguish. We expect small but nonzero activation energies for Na⁺ and Ag⁺ β -aluminas to be measurable with cryogenic cooling. Such cooling is currently beyond our capacities as it would have to entail putting the entire THz beam path under vacuum to operate without cryostat windows. Therefore, we have used heating here.

Reviewer 1 (c): **Figure 4 or K⁺ β -Alumina measurements. Here the picosecond range shown is much shorter than the ones shown before. While lower signal to noise ratio is present, longer times would be beneficial to show whether the TKE signal recorded at 300K does fully decay. Suggestion equal time scale to simulation in Figure 4b.**

This is a great suggestion. The TKE signal as shown is the average of 140 sweeps in time delay at room temperature, and 256 such sweeps at 350 °C, measured over ≈ 7 and 12 hours, respectively. Unfortunately, we did not take data at longer time delays. However, we rule out the possibility that the signal does not eventually fully decay since these are measurements averaged over many shots. An incomplete decay of the signal between pump pulses (1 kHz, and every other pump pulse blocked) would result in a drift of the signal, which was not observed. We show in Figure R1 below the averages over the first halves of the measurements versus those over the second halves of the measurements for both temperatures showing the stability and reproducibility of the response.

In our analysis of the beta''-alumina measurements, we focus on the qualitative existence of a non-instantaneous decay component. This simpler analysis is sufficiently reproduced in simulation and does not rely on the precise kinetics of subsequent decay.

Figure R1 | Verification of absence of drift in TKE measurements of K β''-alumina. (a) Averages over the first half (blue) and second half (orange) of the measurement at 300 K. (b) Averages over the first half (blue) and second half (orange) of the measurement at 350 C. The shaded regions are ± 1 s.e. of the mean.

Reviewer 1 (d): **Different values for simulated attempt frequencies for Na β-Alumina; 1.3 THz in Supporting Information and 1.2 THz for ion pair hopping in the main text. Is an interval or generally an estimation of uncertainties for the frequencies given a more reasonable approach to report those values?**

We are happy to clarify. We have used 1.2 THz in the main text as a proxy for the frequency of a ≈ 5 -meV vibration uncovered by inelastic neutron scattering¹; we are unable to offer more precise uncertainty values as they were not reported in the original study. Additional neutron scattering measurements by the same authors indicate a flat phonon band at ≈ 6 meV² (their Figure 3.18), with some variation across the Brillouin zone, and with linewidth ≈ 1 meV when shown. Our own computational estimate was based on the approximate position of the peak in Figure S10b. While that is a broad peak, we also recognize that fitting can be sensitive to the hyperparameters of where the region of fitting starts and ends within a longer data series and prefer to avoid a complex multi-parameter procedure. Further, there is some variation when the simulated hopping is dis-aggregated by location as done in Ref. 7. We now include this dis-aggregation for the simulated

$\text{Na}_{1.20}$ β -alumina (green curve in Figure S10b) as Figure S10c. The range 1.4 ± 0.2 THz, shaded in grey, yields a conservative estimate of the peak site-filling frequency from simulation. We reproduce the updated Figure S10 as Figure R2 below.

Taking the reported linewidth of literature phonons as 1 meV (≈ 0.25 THz), and ignoring variation across the Brillouin zone, for a difference between reported frequencies to be significant, the vibration should ring for at least 4 ps. That is clearly not happening in the simulation – except for the obviously different vibration at 2.3 THz. In absence of other plausible candidate vibrations, we consider the peak in the Fourier transform of the site filling times indistinguishable from the literature neutron scattering values for all intents and purposes. The significant difference on which we focus is that between the phonon(s) at 5-6 meV and the zone-center optical ones at 2.3 THz, which is highlighted by the simulated composition series in Supplementary Note 6 and Figure S10. It is impressive and reassuring that a classical interatomic potential from the 1970’s yields agreement of this quality. A more detailed investigation of the phononic contributions requires computing a finite-temperature renormalization of the phonon band structure due to strong anharmonicities^{3,4} and is beyond the scope of the present study.

Figure R2 | Attempt frequencies for Na β -alumina from MD simulations. (a) Distributions of times that anti-Beevers-Ross sites spend empty following a Na ion hopping away, simulated at 600 K for $x = 0, 0.01, 0.10$ (blue, orange, and green, respectively) in stoichiometries $\text{Na}_{1+2x}\text{Al}_{11}\text{O}_{17+x}$. (b) Fourier transforms of the distributions in (a), smoothed with a Gaussian filter of 0.1 THz st. dev. Arrows highlight peaks at 2.3 THz and 1.4 THz corresponding to the short-time structure of the distributions in (a). (c) Fourier transforms of the $x = 0.10$ distribution in (a,b) dis-aggregated by the distances from sites to O_i interstitials: “ $r = 0$ ” is immediately adjacent to defects (see Supplementary Note 5 and Figure S7a of [Ref.] for more details), and black is the total distribution. (d) Na vibrational density of states and (e) real part of the optical conductivity σ for the three simulated stoichiometries of Na β -alumina, with arrows highlighting the same frequencies as in (b). (f) Model of the three internal states for mobile ions in Na β -alumina: solo as in the defect-free ($x = 0$) material, part of a mobile-ion pair, and part of a defect cluster. The first two states possess distinct attempt frequencies for hopping, 2.3 THz and 1.4 ± 0.2 THz, respectively, whereas the defect cluster is non-diffusive.

Reviewer 1 (e): Consistency about the uncertainty of activation energy of the decay process in K^+ β -Alumina (Caption Figure 3 and “Picosecond hopping dynamics in β -aluminas” / “Correlations and Memory in Ion Hopping” section); A triplicate measurement in purged atmosphere leads to uncertainties, which could be propagated to a linear fit giving rise to uncertainties of the “guide to the eye” 45 meV line. You give “ ≈ 45 meV” and “ $\approx 40 - 50$ meV”. At least give reason/or state estimated “ ± 5 meV” uncertainty.

In line with the Reviewer’s suggestion, we change the previous guide to a regression fit of the logarithm of lifetime versus inverse temperature, with the activation energy at 41 ± 7 meV for the 200-micron purged sample. We reproduce the updated Figure 3b below as Figure R3.

Figure R3 | Time constants of single-exponential fits to the long-lived TKE component in ambient atmosphere (filled symbols) and dry atmosphere (empty symbols). Error bars are ± 1 s.e. of least-squares fitting. The dashed line is a fit (41 ± 7 meV) to the 200- μ m sample and purged atmosphere.

We have also amended the main text of the manuscript and the caption of Figure 3: “For K^+ β -alumina, this non-oscillatory relaxation accelerates from ≈ 10 ps at 300 K to ≈ 4 ps at 620 K (Figure 3b), with an activation energy of 41 ± 7 meV.” ... “The dashed line at an activation energy of $41(7)$ meV is a fit to the purged-atmosphere measurements.”

For perspective, the activation energies for comparison are NMR relaxation rates, which we fit to the only available literature data by Greenbaum and Strom⁵ at 60 ± 4 meV, and the low-frequency/d.c. conductivity, which has been measured at ≥ 300 meV^{6,7}. The agreement with NMR relaxation is as expected: the NMR activation is an upper bound on the activation of TKE lifetimes. The key difference is that between both NMR and TKE activations and those for d.c. conductivity.

Reviewer 1 (f): Equation 2; (Polarization = Dipoles+ Int(hopping rates)); You correctly describe a non-zero contribution of the integral for hopping in a preferential direction. You describe this to be the case in your experimental setup until the pump drives those. If equilibration at elevated temperatures or any other than a radial temperature gradient is applied an anisotropy of hopping rates might also be present due to temperature, which is most likely negligible but maybe noteworthy.

We expect the contributions from possible gradients to be negligible. First, the probe is centered on the pump when overlap is optimized to achieve the highest possible signal. Second, the pump is $\approx 5x$ larger than the probe, contributing to the uniformity of THz excitation fluence over the area

of sample being probed. We can safely assume a temperature gradient that is both small and radial. Regarding temperature equilibration, we show in Supplementary Note 4 that at least in simulation (a) most of the pump energy is absorbed by mobile ions and subsequently transferred to the lattice, (b) the overall lattice heating is extremely weak, on the order of 1 K. We therefore rule out possible thermopower contributions as well.

Reviewer 1 (g): **Can the observed anisotropy not only be interpreted as the lifetime of an occupied interstitial A site (excited state) comparable to transition state theory? A preferred back hopping into the B site (and not the B' site) might additionally be indicating that not all neighboring B' sites are unoccupied. This would then be a “common” statistical site blocking effect, while “single hops” are probed still an average over a large amount of “single hops” is probed. The lower activation energy of the decay is then only the barrier out of the interstitial A site, and not the B → A site, which should be rate determining (largest activation barrier). This may explain why the Ag⁺ and Na⁺ β-Aluminas do not show a clear activation behavior (as far as I can tell) from the 300 K vs. 620 K measurements (Extended Data Figures 3 and 5) as also stated in Supplementary note 4 more easily. And in the slower ionic conductor K⁺ β-Alumina may however be an activated process for a bit larger energetic barriers for A → B sites (as indicated in longer lifetimes). Maybe decipher A→B and B→A site hops from simulation.**

The Reviewer’s comment touches upon several large topics within the multiscale understanding of ion transport:

1. What is termed “single hops”.
2. The relative importance of interactions between neighboring mobile ions versus the interactions between mobile ions and the host lattice.
3. The existence of a TKE relaxation tail versus the origin of its temperature activation: the “common” statistical site blocking effect.
4. The existence of an atomistic rate-determining step, such as a B→A hop.
5. The smaller magnitude of the TKE activation energy for Na⁺ and Ag⁺ than for K⁺.

We appreciate the opportunity for an in-depth modelling discussion to highlight synergies between the present experimental study and our computational work in Ref. 7. However, we believe that the Reviewer’s remark may concern our study in Ref. 7 more than the present work. Our present manuscript offers a new experimental way to gain insight into the fundamental atomistic nature of transport, including beyond solid-state ion conductors, which merits its publication in *Nature*.

We first follow the Reviewer’s suggestion to dis-aggregate simulations of hopping anisotropy by lattice site, and then respond to the other conceptual points. Figure R4 below shows the simulated anisotropies of hopping by the sites of origin, dis-aggregated by the site from which the hops originate: Beavers-Ross (BR) and anti-Beavers-Ross (aBR), for Na⁺ and K⁺ β-alumina, and for ions that are not bound at defects. The simulated trajectories bear subtle differences, such as a small shift in time delay for the two Na⁺ sites. But for both materials, both sites contribute very similar time traces to the anisotropy. In β-aluminas, this is consistent with significant interactions between mobile ions: the occupancy of neighboring high-energy sites (anti-Beavers-Ross) perturbs the energetics and dynamics of ions at low-energy sites (Beavers-Ross). This is illustrated in Supplementary Note 6 and Figure S10.

We also include in Figure R4ef a counterfactual in which the site-disaggregated TKE responses are distinct. Consider the action of a higher-frequency pump (2.5 THz simulated) on Na⁺ β -alumina. The site-disaggregated time traces of simulated hopping anisotropy differ when pumped at 2.5 THz (orange), but not when pumped at 0.7 THz (teal, experiment, repeating Figure R4ab). Near time zero the 2.5-THz pump drives hopping BR \rightarrow aBR (anisotropy peak \approx 0.5 ps), and at a slightly longer time delay the reverse aBR \rightarrow BR hops take place (anisotropy peak \approx 1-1.5 ps). This distinctive behavior occurs because the higher-frequency 2.5-THz pump excites predominantly the ions in Beavers-Ross sites that are not part of ion pairs at the strong IR-active vibration (Figure S10, $x=0$ in $M_{1+2x}Al_{11}O_{17+x}$). This pathway is of course also present in doped samples, but it is not an appreciable contributor to conductivity (Supplementary Note 6).

Figure R4 | Dependence of the simulated hopping anisotropy in β -aluminas on site type. (ab) hops from Beavers-Ross (BR) sites to anti-Beavers-Ross (aBR) sites. (cd) hops from anti-Beavers-Ross (aBR) sites to Beavers-Ross (BR) sites. (a,c) Na, (b,d) K. Shaded regions represent ± 1 s.e. of the mean. Only sites away from defect clusters are used.

We are grateful to the Reviewer for drawing our attention to this point; its inclusion is both a non-trivial improvement to the manuscript and an additional opportunity to highlight its synergy with Ref. 7. We have added Figure R4 as Figure S12, and amended the Supplementary Note 6:

“We further dis-aggregate the simulated anisotropy of hopping by the site of origin for each hop for Na and K β -aluminas, as shown in Figure S12. Notably, hops from both types of sites show a similar long-lived anisotropy due to the predominant excitation ion-pair hopping by the 0.7-THz pump. When a 2.5-THz pump is simulated (Figure S12ef, orange), the site-specific traces are markedly different. The BR \rightarrow aBR hopping is excited first, followed by aBR \rightarrow BR.”

We now return to the broader conceptual components of the Reviewer’s remark.

What is termed “single hops”. We use the same formalism as in Ref. 7 for distinguishing single hopping events from double hops, which are *sequential* pairs of single hops of the same ion. In Supplementary Note 1 of Ref. 7 we demonstrate that consideration of double hops is necessary to reproduce the absolute magnitudes of ionic conductivities from first principles. Notably, because of the timescale dependence of correlations between hops (Figure 5 and Extended Data Figure 9 of Ref. 7), pairs of single hops even by an ion-pair are not in general equivalent to a double hop. See also the discussion of random walks and the Haven ratio below.

Additionally, we highlight the difference between sampling the whole ensemble over only short-lifetime hops (as in the TKE experiment) versus sampling all possible hopping lifetimes, which is required to reach the ergodic limit and establish constant-valued transport coefficients. Due to limitations of signal to noise resolution, the TKE experiment samples over all sites in the material but only over short hop lifetimes. The value of pairing experiment and simulation (here and in Ref. 7) without extrapolation across decades in timescale is that we can pinpoint which parts of the overall process of transport are being driven experimentally and demonstrate the distinction between ensemble averaging and time averaging.

Site energies, rate-limiting steps, and interpretation as the occupancy of a high-energy site. The treatment of hopping as transitions between low-energy and high-energy sites Reviewer invites is most valid when the (mobile ion)–(host lattice) interactions dominate over (mobile ion)–(mobile ion) interactions. However, we have demonstrated in Supplementary Note 6 and above that ion pairing and (mobile ion)–(mobile ion) interactions are strong or dominant instead. This regime is assured by the practical extrinsic doping ($x \approx 0.1$ in $M_{1+2x}Al_{11}O_{17+x}$) in β -aluminas. Further, in Supplementary Note 5 of Ref. 7, we have calculated the ergodic-limit free energies for BR and a-BR sites in β -aluminas. The differences between such site energies are larger than even the low-frequency (d.c.-limit) activation energies for conduction, even if kinetic barriers are ignored. Therefore, surmounting a singular BR \rightarrow a-BR (B \rightarrow A) barrier cannot be taken as a rate-determining step for conduction. Overall, site energies are not useful descriptors of transport in β -aluminas.

Similarly, we doubt that a singular rate-limiting step for transport in β - and β' -aluminas can be identified. The distributions of survival times are wide-tailed in all materials we have simulated (Supplementary Notes 1 and 4 in Ref. 7), including for the defect clusters in β -aluminas (Supplementary Note 5 of Ref. 7). Overall, the strength of (mobile ion)–(mobile ion) interactions,

the dispersion in site energies and rates of hopping, and strong correlations between successive hops all invite an understanding of transport that harkens to (nanoconfined) complex fluids, which is what we advance. In such a model, a more useful basis set and predictor of short-time dynamics is the “state” of a mobile ion (schematized in Figure S10f, ion-pair vs unpaired) rather than its location (BR vs a-BR). Here, a double hop as defined above involves an ion-pair hop, a re-association forming a different ion-pair, and a second hop within the second ion-pair. In somewhat oversimplified terms, the memory, mutual information, and correlations can be most directly embodied in the re-association step. Comparing experiment and simulation, our pump evidently suffices to move a few ion pairs between lattice sites, but not to dis-associate them. Overall, this situation is in line with the concepts of Funke⁸ and of Song *et al.*⁹ that we extend in Ref. 7. A conceptually similar model has been developed for complex fluids such as battery electrolytes¹⁰ exhibiting vehicular and non-vehicular transport. This effect is likely to generalize to ion conductors with non-equivalent lattice sites (or chemical environments) and non-negligible populations of the nominally high-energy sites, i.e., most modern ion conductors.

Activation energies for TKE in Na⁺ and Ag⁺ β-aluminas vs K⁺ β-alumina. The activation energies for Na⁺ and Ag⁺ β-aluminas are expected to be smaller than for K⁺ β-alumina, but nonetheless distinct from zero. Figure R4 above also demonstrates that the pump responses of K⁺ and Na⁺ β-aluminas are dominated by ion pairs and arise from the same dynamics. We discuss additional complications arising from the presence of bound defect clusters in Supplementary Note 6; defect clusters are most responsive and most dissociative for Na⁺ and Ag⁺, whereas they are most stable for K⁺ (Supplementary Note 5 of Ref. 7).

An upper bound on the expected activations for TKE relaxation can be found employing time-temperature superposition¹¹ from the lowest-temperature regimes of other measurements, such as NMR relaxation or microwave conductivity. We include literature examples of activation energies in Table R1 below. A similar summary of the drop in activation energies with increasing frequency of measurement is also provided by Kamishima¹² for Ag⁺ β-alumina (their Figure 6), although they do not distinguish temperature regimes when multiple ones exist for the same technique (NMR). In all our samples, our TKE measurements yield an energy lower than literature NMR or GHz ones, which constitutes agreement with literature. From Table R1 (included as Table S1), it is evident that activations for Na⁺ and Ag⁺ β-aluminas would be weaker than in K⁺ β-alumina. Finally, the equivalent activation energies for β''-aluminas in their low-temperature regime (only Na⁺ ever measured, Table R1) are similarly expected to be smaller than even our measured activation for K⁺ β-alumina, and we pursued a qualitative comparison between regimes instead.

Table R1: high-frequency activation energies from NMR relaxation rates and high-frequency conductivity measurements (denoted as GHz). The lack of an uncertainty value for Ref. 16 reflects that the value is taken from Ref. 16 directly rather than from our own fitting, and uncertainty is not reported there. The lack of uncertainty values for Barker *et al.*¹⁷ reflects the lack of clarity on the co-variation of data reported there (see their Figure 5), or our use of their reported values. The lack of uncertainty values for Funke¹⁹ reflects the paucity of data points. Otherwise, our fitting uncertainty values are ±1 s.e.

Material	Method	Frequency, MHz	Low T E_A , meV	High T E_A , meV	Ref.
Na β-alumina, melt	NMR	17.2	30±2	181±15	[13]

Na β -alumina, melt	NMR	25.5	40 \pm 2	105 \pm 4	[13]
Ag β -alumina	NMR	400	28 \pm 1	98 \pm 5	[14]
Na β -alumina, melt	NMR	21	42 \pm 2	135 \pm 4	[15]
Na β -alumina	NMR	5.2	39	N/A	[16]
K β -alumina	NMR	11.1	60 \pm 4	N/A	[5]
Na β -alumina, melt	GHz	1200	41	118	[17]
Na β -alumina, melt	GHz	24000	66		[17]
Ag β -alumina	GHz	380	24, 54	145	[17]
Ag β -alumina	GHz	24000	N/A	110	[17]
K β -alumina	GHz	1200	N/A	192	[17]
K β -alumina	GHz	24000	N/A	116	[17]
Na β'' -alumina	NMR	39.6	39 \pm 5, 98 \pm 7	252 \pm 9	[18]
Na β'' -alumina	GHz	18000	30	199	[19]
Na β'' -alumina	GHz	60000	29	N/A	[19]

The interpretation of temperature activation of the TKE tail as a single energy. Due to the strength of (mobile-ion)–(mobile-ion) interactions and preference for ion-pair hopping, a more direct analogy is that with Funke’s concept of mismatch and relaxation (Ref. 6 in the original manuscript), which invokes kinetic competition between returning and non-returning hop rates. We have explored this parallel further in Ref. 7. The present manuscript provides an experimental verification of this cornerstone idea. There are three challenges to a simplified interpretation in terms of a site energy, which corresponds most directly to the case we discuss in Figure S9a.

First, since ion pairs rather than isolated hops dominate the TKE responses in the simulated β -aluminas (Figure R4), TKE activations cannot be interpreted as site energies or as purely kinetic barriers for isolated or “single” hops. That said, the bottlenecks caused by the host lattice slow down both ion-pair hopping and unpaired-ion hopping. Since K^+ is larger than both Na^+ and Ag^+ , both (mobile ion)–(host lattice) interactions and (mobile-ion)–(mobile-ion) repulsions are the strongest in K^+ β -alumina. Overall, as discussed in Ref. 7, the activation energy arises from a balance between (mobile ion)–(host lattice) interaction, and (mobile-ion)–(mobile-ion) repulsions. Both are non-negligible, and the activations should not be simplified further. An interpretation of TKE activation as a single-hop site-energy would correspond to a regime where (mobile ion)–(host lattice) interactions dominate, which we illustrate above in Figure R4ef for the 2.5-THz pump and non-ion-pair hops. That is not the case for the 0.7-THz experimental pump. If we were to invoke site energies, we would have to invoke at least two distinct ones to accommodate two hopping mechanisms, ion-pair and unpaired-ion. We recognize that it will always remain possible to parameterize a hopping model with a large enough set of site energies, repulsion energies, or frequencies. Our study opens the opportunity to verify whether such models remain accurate at the picosecond timescale.

Second, The TKE signal arises within the honeycomb lattice from a kinetic competition. The numerator of the anisotropy signal is the difference between hopping rates parallel and perpendicular to the pump polarization, which in the trigonal lattice conveniently correspond to returning and non-returning hops. The kinetic competition inherent in the TKE measurement lends a direct parallel to Funke’s model and distinguishes the TKE measurement from others such as

NMR. Unfortunately, the measurement of a difference between two rates is not sufficient to measure either one individually.

Third, site energies are significantly inhomogeneous throughout a material. An interesting question beyond the scope of the present study is whether Markov-state models from kinetic Monte-Carlo (kMC) simulation, e.g., most recently Deng *et al.*²⁰, can reproduce frequency dependences of conductivities and correlation factors in the atomistic limit – or if their predictions only apply in the ergodic limit despite their parameterization from first principles. Both our Ref. 7 and Deng *et al.* demonstrate sufficient variation in individual site free energies that a “high-energy site” and a “low-energy site” can hardly be identified in either case. This and the more general question of how Markov-state models parameterized from single-hop transition barriers correspond to experimental measurements of conductivity and its activation energy across timescales are intriguing directions for future research. We touch on this below in the discussion of Markovianity, but all of this is beyond the scope of the present study.

‘The “common” statistical site blocking effect’ and the existence of the TKE tail versus its activation. The Reviewer’s reference to site blocking is closest to the higher-frequency pumping simulation (orange in Figure R4ef above), where non-ion-pair excitations are pumped. But that is not the case in general, and not the case for the experimental excitation of predominantly ion-pair hopping in β -aluminas with the 0.7-THz pump. The Reviewer’s remarks, which we assume arise from the solid-state ionics community, have given us an opportunity to clarify our models and examples. The possible confusion here is between the existence of a TKE tail, and its activation.

The activation energy for the TKE tail convolves both the absolute values and the activation energies for the competing rates making up the signal. In general, the competing activations do not have to be equal. Further, additional relaxation mechanisms are possible (per Funke’s work). In β -aluminas and β'' -aluminas, host-lattice dynamics can be assumed to be minimal, which assists us in isolating the hopping component of TKE – but does not rule out ion-pairing. For example, both ions within a coupled ion pair can be driven by the pump in the same direction – which is not equivalent to a double hop of one ion (see above under “what is a single hop” and “site energies, rate-limiting steps, etc”). With the ion-pair pathway being dominant, the memory component is most closely embodied in the association of an ion with one of its multiple neighbors.

However, our information-theory framework generalizes beyond ion pairing and β -aluminas to more modern materials where, e.g., a polyanion flip could turn site A into site B on its own timescale, or to complex fluids where the exchange of coordination environments can mimic solid-state hopping as a non-vehicular mode of transport. All these relaxation pathways could contribute to TKE activation outside of the “statistical site-blocking effect”. We chose to formulate the discussion in Supplementary Note 5 in a way that does not rely on knowledge of specific mechanisms, although we use a well-constrained model system in our specific experiment and simulation to distinguish those. This may have created confusion. We now add specific statements to match the general cases in Figure S9 to specific examples in our model materials:

In the discussion of Figure S9a: *“This case corresponds most closely to the higher-frequency pump of Na β -alumina discussed below (Supplementary Note 6) that drives hopping by ions that are not part of ion pairs and predominantly relax by returning hops.”*

In the discussion of Figure S9b: “*An example of this case is the experimental pumping of β -aluminas; due to the predominance of an ion-pair pathway of hopping (Figure S10 below), both rates are non-negligible. Within ion-pair hopping, the memory component can be conceptualized as the (non-)random association of an ion, e.g. at site A, with one of its neighbors.*”

Reviewer 1 (h):

Equations should be numbered (and/or fully included into the text.)

Extended Data Figure 2c; Is the normalization regime correct?

Extended Data Figure 3a; Temperature is given as 620K, maybe 300K?

Extended Data Figure 5e; Temperature is given as 3000K, maybe 300K?

Figure 3 or temperature dependent measurements. Those do not overlap with the ones depicted in extended data Figure 2 and 4. It would be beneficial, to give the atmosphere used in the caption of every Figure containing TKE signals.

We are grateful to the Reviewer for these detailed comments that improve the manuscript.

- All equations are now numbered in the main text and supplementary information.
- In Extended Data Figure (EDF) 2c and all other panels in EDF 2, the normalization is applied at a time delay that is large enough for the instantaneous component to fully decay. The normalization is correct as far as we can tell. There are two reasons the normalization for the K^+ samples in EDF 2c may look different from that for the Na^+ (panels a-b) and Ag^+ (panels e-f) samples. First, the vibrational frequencies excited in the K^+ sample are higher than in the other two (Figure 2), which leads to their phase being more washed out in the thick samples due to pump-probe walk-off, which in turn leads to the thin sample having a much more pronounced vibrational structure and a larger short-time peak relative to the thick sample when normalized. Second, in the K^+ sample, the “tail” is weaker than in the Na^+ sample, making the vibrational structure at short time delays relatively stronger upon normalization; the absorption is strongest and the difference between thick and thin samples is minimized in the Ag^+ samples due to the best overlap between the pump and the ionic vibrations. Both these factors contribute to the distinctive look of EDF 2c, which is nonetheless correct.
- EDF 3a and EDF 5e indeed show data measured at 300 K and are now re-labeled as such. We are grateful for the pointers and corrections.
- Figure 3a shows a temperature series taken in ambient atmosphere; chronologically, we took this series first, before upgrading the system for moisture-free measurements. Given the need for nearly a week of consecutive measurement time, we have only repeated the endpoint temperatures since then. The lifetimes shown in Figure 3b and traces of the birefringence arising from moisture shown in Supplementary Figure S1. Despite the moisture, Figure 3a is our largest fully self-consistent dataset, and we believe its use to be appropriate.

Reviewer 1 (i): **It may be helpful to add this reference (End of Section “Probing correlated Hopping” or Supplementary Note 6) of a theoretical study excitation of certain vibrational modes relevant for ion diffusion, as well as a differing (LAMMPS) temperature of host lattice and mobile ions (Supplementary Note 4): Gordiz et al. Enhancement of ion diffusion by targeted phonon excitation, Cell Rep. Phys. Sci. 2021, 2, 100431; DOI: 10.1016/j.xcrp.2021.100431**

In line with the reviewer's suggestion, we have added a reference to this paper in Supplementary Note 4: "We note that our computational approach of simulating experimental impulsive pumps is distinct from the computational partitioning of energy into pre-selected vibrational modes in a recent study by Gordiz *et al.* [Ref. Gordiz *et al.*]"

Important differences between our study and that by Gordiz *et al.* include (but are not limited to):

Gordiz *et al.* use strong thermostating to add energy to a select few modes. From their methods section: "*the addition of energy to the top five modes was complimented [sic] by a uniform reduction in the kinetic energy of all other modes in the system,*" which is equivalent to thermostating the simulation. Furthermore, Gordiz *et al.* apply multiple such partitions to the same ion at time intervals (5-20 fs) much shorter than either hopping or periods of any phonons.

If we attempt to translate such velocity additions to an experiment, they correspond to the absorption of pump photons (or an ultrafast heat pump). The absorption of multiple photons per hop (each carrying ≈ 40 meV at ≈ 10 THz) to mimic the simulation by Gordiz *et al.*, requires a pump power at least an order of magnitude above our experiment and simulation, even before accounting for polarization or absorption by non-hopping ions at the same frequencies. Overall, the rapid application of energy in the study by Gordiz *et al.* likely overwhelms the correlation effects that are key to practical experimental studies. These correlation effects are the focus of our study.

By contrast, in our simulations, the transiently differing temperatures of the lattice and the mobile ions arise from physical limitations of vibrational heat transfer following a pump that approximates experiment (Supplementary Note 4, Figure S6). Our choice of model system ensures that the terahertz pumping (both experimental and simulated) is near-resonant with mobile-ion motions, but not with the host lattice. In our study, this results in the pump energy being deposited firstly in the mobile ions, which LAMMPS interprets as a transient uptick in their temperature. Our simulated pumps corresponding to experimental fluences raise mobile-ion temperature by at most ≈ 10 degrees, and only transiently so. In our simulation, the overall lattice temperature rises by at most ≈ 1 degree when the mobile ions thermalize with it in the constant-energy (NVE) ensemble.

In terms of extracting fundamental insight or reduction to practice, Gordiz *et al.* do not involve a phonon dispersion or distinguish between modes coupling mobile ions to host lattice versus modes that carry predominantly mobile-ion displacements, which we do empirically in Supplementary Note 6. (This differs from the projection onto NEB hopping coordinates that they employ.)

Reviewer 1 (j): Considering Figure 1b and the discussion of a random walk: The concept of a random walk is only ever defined for a sequence of ionic jumps. Those can include correlation effects, which distinguish a random walk from a "true random walk" and are associated to the correlation factors (or the Haven ratio in solid state ionics) [H. Mehrer, Diffusion in solids, 2006]. I would encourage more carefully phrasing of paragraph 2 in the "models of Ion Transport" Section.

We thank the Reviewer for this comment. The Reviewer's comment brings to the fore differing approaches to random-walk models across the fields of science studying transport. Solid-state ionics literature focuses on the ensemble of ions as they comprise a macroscopic diffusion

coefficient. We conjecture that this has historically been the case due to the inaccessibility of experimental probes of atomistic hopping outcomes. In absence of such probes, macroscopic-derived models have been predicated on each individual hop being Markovian. Markovianity, when applied to a process from the atoms up as we do here, implies the absence of memory between steps of a process. We appreciate the Reviewer pointing us to Mehrer's book, where this assumption is clearly stated (Section 4.1). We raise this point above with regard to Monte-Carlo methods. The assumption of the Markovianity of each hop stems from the maximum entropy principle: in absence of detailed atomistic information (which our study offers), one must assume that hops are uncorrelated, i.e., each hop evolves the full entropy of the walk in its information theory meaning. And so it remains possible for a transport process to be characterized by non-unity correlation factors at the macroscopic level even if the atomistic hopping process is Markovian and random. For example, if for a dilute vacancy there is no telling which of its neighbors will swap with it next, the self-correlation factor for ions can still be non-unity. But in this common exemplar case where each hop is Markovian, TKE would yield no tail, which in our experiments only happens in the high-temperature regime in beta-doubleprime alumina (Figure 4).

The assumption of Markovianity has been strained over the past decade by, e.g., many simulations of concerted hopping dynamics in modern ion conductors^{21,22}. Here and in our Ref. 7, we draw inspiration from fields beyond solid-state ionics that characterize the (non-)randomness of a walk using generating functions (see, e.g., I.M. Sokolov and J. Klafter, *First Steps in Random Walks*, 2011, main text Ref. 19). Two distributions generate the “non-randomness” of a process: (A) the waiting times between hops, as in a semi-Markovian continuous-time random walk, and (B) mutual information between the directions of successive hops and the waiting times between them. The distributions (A) and (B) fully define transport across all timescales. The functional form of (A) and (B) accounts for the dispersion at fast (ps-ns) timescales but is often overlooked in solid-state ionics due to the lack of experimental probes that track the outcomes of individual hops.

For a simple example of (A), an exponential distribution is the maximum-entropy distribution on $[0, \infty)$ subject to the constraint of a finite first moment. A macroscopic measurement only carries first-moment (ensemble average) information, and thus only an exponential distribution can be parameterized from macroscopic measurements. Further, an ergodic-regime measurement convolutes (A) and (B). Since a macroscopic measurement can yield the same result from many pairs of (A) and (B), the maximum entropy principle dictates that *in absence of higher-moment information* the waiting times between steps must be modelled as exponentially distributed.

However, if waiting times are not exponential, or mutual information between direction and waiting times exists, additional parameters are necessary to define the distributions, and thus more information, i.e., memory, is needed to *construct* the walk^{23,24} and model transport across all timescales. In β - and β'' -aluminas, the distributions of waiting times between hops are indeed wide-tailed (Supplementary Note 4 of Ref. 7), as are the periods that sites stay empty before a distinct ion arrives, which renders both ion and vacancy “walks” at best semi-Markovian even before directionality is considered. Figure 1b in the main text generalizes our bottom-up approach.

The two approaches converge in the ergodic limit; how much the Markovianity assumption remains valid for a true multi scale understanding of condensed-phase transport is the subject of future studies. In the present work, we offer early experimental evidence to verify when ion

transport (the hopping process) in the condensed phases can be considered correlated (non-Markovian) or uncorrelated (random-walk, Markovian) – a distinction that does not require a macroscopic-length sequence of hops to describe. Our work puts light between whether each hop is Markovian versus whether the ergodic ensemble-average transport exhibits non-unity correlation factors when measured macroscopically. The high potential for confusion if we were to mention correlation factors and the Haven ratio is the reason for our avoidance of the terms here. Our study pertains to correlations between hops on the picosecond scale. While our approach differs from that of traditional solid-state ionics, it carries the possibility of developing an atomistically correct multiscale bottom-up understanding of transport in condensed phases. Here and in Ref. 7, we invite a treatment of transport in fast-ion conductors in the same framework as that of complex fluids, where non-Markovian dynamics can exist even in absence of thermal activation or a clear-cut hopping process. For example, we show (Supplementary Note 9 of Ref. 7) that simulated high-frequency impedance in ion transport is consistent with the fractional Fokker-Planck equation rather than with the “normal” one due to non-ergodic dispersion.

As an aside, it will always remain possible to parameterize a model of transport using Markovian hops, but the number of empirical parameters (e.g. site energies, ion-ion repulsions, Lidiard’s five frequencies, attempt frequencies, etc) required to do so is large enough to wiggle the proverbial elephant’s tail. Due again to lack of experimental probes on the single-hop timescale, it has generally not been possible to verify whether such macroscopic-derived models yield a true multiscale picture of transport. For example, such models leave the attempt frequency as a floating or approximate parameter, which is why we comment on the physical nature of the attempt frequency in this manuscript. An illustrative example is that by Meyer *et al.*,²⁵ which attempts to reproduce a frequency dependence of AC conductivity in beta/beta-doubleprime aluminas, but their parameterization of attempt frequencies is unphysical by an order of magnitude. By pinpointing the mechanism-resolved vibrational attempt frequencies, we reduce the number of available fudge factors by at least one.

Nonetheless, it behooves us to clarify in the introduction that we are referring to transport as the hopping process itself. In line with the Reviewer’s suggestion, we make the first two paragraphs of the introduction more precise: *“Linking the mechanistic features of ion transport at the atomic level to the collective descriptors of macroscopic transport within a multi-scale model, or as measured in a device, yields opportunities to design novel processes and applications [Refs]. Fast-ion transport in the solid state commands particular attention due to its importance in energy and information technologies such as solid-state batteries and non-volatile memory [Refs]. Solid-state ion transport is composed of rapid translations between lattice sites, called hops, which constitute the fundamental steps of diffusion and conduction (Figure 1a).”*

However, the correspondence between atomistic and macroscopic regimes in ion transport is often not well characterized: while nanoscale ionic transport is heterogeneous, dispersive, and non-ergodic [Refs], correlation-sensitive probes of atomistic paths analogous to single particle tracking in biophysics do not exist. Without such information, the maximum-entropy principle compels models of transport constructed from macroscopic measurements to assume a Markovian random walk [Refs]. From the standpoint of information theory [Ref], the full overall entropy of transport is evolved at every step of this memory-free process. Traditionally equated to hops in

solid-state transport, these steps originate randomly with an attempt frequency ν_0 and succeed with a probability determined by the Gibbs free energy of a transition state [Refs].”

Notably, these statements remain true even if a non-unity tracer correlation factor characterizes the overall sequence of an ion’s jumps. From the bottom up, such a factor would be fully defined by the distributions (A) and (B). An *a priori* known average-valued tracer (or self-) correlation factor constrains the entropy of a walk, but all of that entropy is still evolved at every step.

Reviewer 2 writes (a): **Overall, I find the manuscript and the conclusions rather technical.**

We contend that the manuscript has broad interest/impact and is therefore not overly technical in connecting, for the first time, solid-state ionics, non-linear optics, and the thermodynamics of information embodied in the correlations of nanoscale transport. Practitioners of all these fields within the broad readership of *Nature* will find this manuscript of interest. Our experimental work extends the field of information thermodynamics to transport processes, where the correlations are not only positional, but also directional and temporal. At its core, ours is a manuscript about the fundamental nature of mass transport in condensed phases (solids and fluids) being revealed by a novel method of probing it.

The experimental and computational methods we use are applicable to studying mass transport processes in solids and fluids at the fastest possible timescales, which will pave the way to exploiting nonlinear, stochastic, and path-dependent phenomena in devices. We have indicated as much by including a broad introduction to the manuscript and referencing studies from fields beyond batteries (Refs. 3, 8-10, 17-19, 23-24). We would have cited additional works on nanoconfined fluids if not for the 50-reference limit, so we chose to cite a review (Ref. 9).

We reproduce our introduction, with the clarifications made in response to Reviewer 1: “...*The correspondence between atomistic and macroscopic regimes in ion transport is often not well characterized: while nanoscale ionic transport is heterogeneous, dispersive and non-ergodic [Refs], correlation-sensitive probes of atomistic paths analogous to single particle tracking in biophysics do not exist. Without such information, the maximum-entropy principle compels models of transport constructed from macroscopic measurements to assume a Markovian random walk [Refs]. From the standpoint of information theory [Refs], the full entropy of transport is evolved at every step of this memory-free process. Traditionally equated to hops in solid-state transport, these steps originate randomly with an attempt frequency ν_0 , and succeed with a probability determined by the Gibbs free energy of a transition state [Refs].*

By contrast, models of correlated transport allow for the macroscopic process of conduction to consist of multiple interconnected steps. Examples include memory kernels in generalized master equations [Refs], Burnett-order nonlinear hydrodynamics [Refs], or kinetic competition [Refs]. In such models, the state of the material at time zero (Figure 1b, green) influences transport dynamics over some non-negligible timescale (e.g., t_1 in Figure 1b). ... Understanding correlation effects in transport remains necessary to predict practical performance of ionic conductors from the atoms up [Refs], and to exploit nonlinear nanoscale transport phenomena [Refs] in devices. Here, we study the correspondence between the thermodynamics of ion transport and those of information.”

Reviewer 2 (b): **prior results in the literature (cited by the authors) have already established that THz pulses can trigger ionic hopping processes in solid battery electrolytes such as beta-aluminas (especially [41] Minami et al. “Macroscopic Ionic Flow in a Superionic Conductor Na+ β -Alumina Driven by Single-Cycle Terahertz Pulses”, Phys. Rev. Lett. 2020), so this aspect of the results is not in itself a key scientific advance.**

We recognize such previous work, but it should be noted that we do not claim that the excitation of ionic hops is the focus nor the novelty of the paper. Instead, it is our unique ability to track

hopping outcomes and derive transferable insights about the fundamental nature of transport in condensed phases that distinguishes our study.

However, unlike Minami *et al.*, we demonstrate that the signal we are measuring indeed arises from bulk ionic hopping by eliminating all other possible contributions. For example, our measurement is done without contacts, which introduce electrons. An electron (assuming unity effective mass) is 4.2×10^4 times lighter than a sodium ion. The presence of electrons at contacts as in the study by Minami *et al.* requires that the pump field at the contacts is that much smaller than in the ionic conductor so that the measured response is not dominated by intraband excitations of electrons at the contacts; this is not ascertained in their study. Furthermore, to model their results, Minami *et al.* use an unexplained factor $\beta=62$ in their equation (1). That leads them to conclude that ions move by nanometer-scale distances (i.e., across multiple unit cells) in response to the pump, which would require the ions to move at velocities comparable to or exceeding the speed of sound in the material. By comparison, we carry out extensive atomistic simulations expressly eliminating such arbitrary factors and simulating a pump strength that is equal to experiment. We obtain small average displacements in the pumped simulations (Supplementary Note 4): “*The simulated coherent displacements of the mobile-ion center of mass are between 5-15 picometers during the application of the pulse, in line with terahertz-frequency pump-probe studies in solid-state materials.*”

In atomistic terms, the terahertz pump is very weak (≈ 3 mV/Å inside the sample), and pump-driven ion hops are rare; that is why we have resorted to extremely large-scale simulations. We also note that our THz transmission results (our Figure S5) disagree with those of Minami *et al.* (their Figure 3b): we use stronger peak fields, but do not observe any intensity dependence of THz transmission in any of the β -aluminas, either in terms of bleaching the absorption or softening of the vibrational mode (peak frequency), which is key to their argument. Our extensive simulations support our results. We have already noted this in the Supplementary Note 3 adjacent to our Figure S5: “*Unlike in recent work [Ref. Minami et al.], no field-strength-dependent shifting or bleaching of any features are observed over multiple measurements of each sample*”. Our results also do not point to the existence of a threshold (compare our Figure S3 with their Figure 2). Overall, we are baffled by their claim to have measured an ionic current across a β -alumina sample with a “conventional ammeter” without any mention of the ion-permeability and air stability of their contacts. A simpler explanation could be that a larger fraction than one in 4.2×10^4 pump photons reaches the contacts and interacts with electrons there, giving rise to the recorded currents. We could continue with a proper rebuttal to their paper, but that is beyond the scope of our work.

Importantly, our work does not rely on the TKE response of ionic conductors being intensity dependent. Instead, one of the conceptual advances in our work is our use of the path-dependence rather than the intensity-dependence of nonlinear optical tools to explore correlations in ionic motions as their transient anisotropy. We separate the excitation and the probe in the time domain by using pairs of time-delayed pulses, which yields specifically information on the direction of incoherent ionic hopping motions following the pump. In doing so, our work is qualitatively novel.

Reviewer 2 (c): **The beta aluminas have been extensively studied in past decades so the choice of materials does not bring much novelty.**

We want to stress that the beta-aluminas were not investigated for novelty, but for the extensive body of literature documenting their various properties as an exemplar model system, and in particular measuring their conductivities at a variety of frequencies. These are directly relevant to our unique experiments and novel concepts. For example, the absence of rotational modes and the trigonal lattice for ionic hopping in β -aluminas enable the detailed interpretation of experimental observables in our study. We further note that *Nature* routinely publishes studies on well-established model systems. Here are two recent examples:

- <https://www.nature.com/articles/s41586-021-03509-z>.
- <https://www.nature.com/articles/s41586-021-03646-5>.

Reviewer 2 (d): a broader question arises concerning the relevance of the methods/results to solid electrolytes for battery applications. The material response detected in these ultrafast nonlinear optical experiments is under intense ultrafast THz pulses that bring the samples very far from equilibrium. It is fairly unclear whether the state of the material and its transient response to such excitation is truly relevant to battery applications, a connection the authors attempt to make.

Regarding the relevance of our results to battery applications, we have for the first time measured a path-dependent, viscoelastic effect in a solid-state system directly relevant to solid electrolytes for battery applications. Such an effect is not measurable macroscopically and requires a nonlinear technique. Because of that requirement, dynamic correlations, that is, memory effects, have so far remained precisely the missing piece for the rational design of materials and batteries from first principles. We have outlined the theory for this in our computational work focusing on longer timescales (Ref. 7). The current manuscript is the experimental observation of the phenomena underpinning our theoretical framework (Ref. 7).

With modifications for Onsager coefficients, the same concepts, tools, and methods as in the present study apply to engineering ion transport in solids and fluids, in and beyond the context of battery applications. In atomistic terms, the applied pumping THz fields are quite weak. When accounting for a front-surface reflection, the peak pump field inside the sample is about 300 kV/cm, which is 3 mV/Å. For comparison, this is lower than the commonly used thresholds for interatomic forces in static density-functional theory calculations, 5-10 mV/Å. In atomistic terms, our fields are within the noise. This is the reason for our use of extremely large simulations: at these field strengths, pump-induced hopping remains rare and representative of transport at, e.g., battery operating conditions.

Furthermore, as noted above, our measurements and analysis do not rely on the strength of the field beyond having a birefringence signal to detect. This is illustrated in Figure S3: there are no changes of regime between zero and maximum applied pump fields. The dynamics that we measure do not rely on an effect induced by an extraordinarily strong pump being different from those at smaller applied fields. Our results are therefore generally applicable. The processes studied here are indeed those that occur under battery operating conditions, and not highly non-equilibrium or exotic responses.

Finally, we believe that *Nature* remains a place for interdisciplinary fundamental science. We reiterate that our study concerns the fundamental nature of transport in condensed phases, and just

happens to use a battery material as a model system due to the plethora of available literature characterizing transport in the battery and solid-state ionics communities. The present study is more similar in its scope to the following than to others of narrower application to batteries:

- Studies demonstrating experimentally the connections between thermodynamics and information theory: Collin *et al.*²⁶ (Nature, 2005) and Bérut *et al.*²⁷ (Nature, 2012). Our study offers a path to probing transport processes and information thermodynamics effects at the atomistic level, where even optical tweezers cannot reach.
- Studies probing the differences between transport properties at the nanoscale and at the macroscale: Siria *et al.*²⁸ (Nature, 2013), Secchi *et al.*²⁹ (Nature, 2016), Mousterde *et al.*³⁰ (Nature, 2019), and Kavokine *et al.*³¹ (Nature, 2022). Our study draws and exploits a parallel between the path dependences in non-linear hydrodynamics (which likely underpin the exotic behaviors of nano-confined systems in the referenced studies) and non-linear optics. We provide both an additional example in a practically relevant context and a new way of probing such effects.
- Studies of the fundamental properties of common substances such as water: Yang *et al.*³² (Nature, 2021) and Mason *et al.*³³ (Nature, 2021).

Reviewer 2 writes: **the details of response seem challenging to interpret reliably. For instance, it is not discussed how much the signal detected depends on details of the THz pulse shape. I also noted that some of the time traces exhibited time structure on fast scale (e.g. sharper/shorter features) than the pump itself, which was not explained (Fig 2, Extended data Fig 1).**

We are happy to provide further clarification. We do not excite any vibrations that are outside of the spectral range of the pump. While the peak of the pump power is at 0.7 THz, the pump spectrum extends to 3.5 THz, as can be seen in Figure 2b (green). We do not observe any vibrations above 3.0 THz. The 3.0 THz vibration in β -aluminas is known to be very strong³⁴. In principle, through nonlinear phononics³⁵, substantially higher frequency vibrations could be excited³⁶, but we have explicitly ruled out such effects in this study.

The lithium niobate THz source offers at best limited opportunities to modify the pulse shape without losing peak field strength. We have chosen to optimize our experiment for the strongest possible signals. The influence of the frequency content of the THz pulse is discussed in part in Figure 4c and Supplementary Note 6 as it pertains to finding attempt frequencies. As this study analyzes primarily the long-time component of birefringence response rather than the short-time coherent vibrational response, further studies analyzing, e.g., the relative phase of frequency components of the pump, are substantially beyond the scope of this work. We agree with the referee this would be an interesting avenue for further study.

Reviewer 2 (e): **The separation of fast/slow response and the fitting procedure illustrated in extended data Fig 3,4,5 is not fully justified/detailed and brings uncertainty to the analysis. Some data look rather noisy (also extended data Fig 7).**

Fitting multi-component signals that (a) are sinusoidal in nature, (b) have a nontrivial rise shape, and (c) decay with potentially different time constants is notoriously difficult. We have built up our fitting procedure from studies of control samples (Figures S1-S3) and grounded it in

literature^{37–39}. At the same time, we already acknowledge that our fitting procedures are imperfect, e.g., in the caption to Extended Data Figure 3: “*The strong vibrational component in the thick-sample signal (a-d) precluded unambiguous fitting at short times, so the exponential component was fit to the long-time part of the signal.*” We will be happy to explore alternative superior fitting procedures that the reviewer is specifically aware of and that can be reliably parameterized.

The frequency content in the spectra of residuals in Extended Data Figures 3-5 and 7 looks noisy because it comes from the residuals to a fit; that there is an interpretable vibrational structure there at all, and furthermore one that is consistent across all measured materials, is already remarkable. Finally, we stress that the main argument of the paper concerns the non-oscillatory relaxation tail rather than the short-time coherent vibrational structure and does not rely on either of these figures.

Reviewer 2 (f): The MD simulations are performed using simple potentials re-used from the literature.

It is broadly recognized that any interatomic potentials-based model is only as good as its validation versus experiment. The potentials we use have withstood the test of time and we contend that they are not “simple” but very effective. Indeed, we have recently used the same potentials for a larger-scale simulation study of the same materials (now published in *Nature Materials*, Ref. 7 in the manuscript), and benchmarked their performance versus frequency spectra of a.c. conductivity (Figure 3 in Ref. 7) – which is a high level of quality assurance, to our knowledge surpassing any other molecular dynamics simulation study.

While it is now possible to train bespoke neural-network potentials from ab initio accuracy, none of them still scale as well as the classical potentials we use here. We would not have been able to conduct simulations of the size required to match experiments here – 4 million mobile ions per material per temperature taken together – with more complex potentials. The high ionicity of the materials in this study further justifies the use of classical potentials. Further, similar simulations of water^{40–42}, a system with more significant polarization effects, confirm that classical potentials-based methods are sufficient and appropriate for our use in studying ionic motions.

Reviewer 2 (g): It is not clear that these would be appropriate for the intensely excited sample. For example, the electric field could modify the height of the potential barrier, changing the potential and ion dynamics.

We reiterate that the THz pump field is weak, not strong, in atomistic terms: ≈ 3 mV/Å inside the samples. Furthermore, the bulk of our study concerns not the ion dynamics while the pump is applied, but the subsequent ion dynamics when the pump pulse has already exited the sample. If there were indeed a changed potential energy landscape after the pump is turned off, then that is definitionally a memory effect, much like the subject of the present study.

Reviewer 2 (h): The section reporting sample preparation in supplement is very brief. I found the degree of characterization of samples reported insufficient to establish the full ion exchange (in case of Ag, K), or the crystal quality (before and after intense THz pump experiments).

Again, we point out that the pump fields are weak, and the total amount of energy deposited in the samples by the pump is small. We simulate the temperature rise in the samples to be on the order of 1 degree Celsius (Supplementary Note 4, Figure S5). We have carried out extensive longevity testing as shown in Figure S4. The samples were re-used numerous times, and we never saw any indications of damage from THz pump-probe experiments. For example, the moisture-free measurements of the time constants for which we report in Figure 3b were carried out in three separate sessions over the course of two calendar months. We further note that the TKE experiment is bulk-sensitive rather than interface-, disorder-, or impurity-sensitive, and we have not seen any signatures, either crystallographic or vibrational or otherwise, that would contradict our reported characterization. Finally, the c lattice parameter and completion of mass change are the standard reporting metrics for ion exchange in β -aluminas. See for example the reporting by Kamishima *et al.*^{12,14,43-47}, who provided the crystals we have used.

Additionally, we have characterized the thin polycrystalline K β "-alumina sample following TKE measurements and additional high-intensity 800 nm illumination to complement the stronger 800nm pulses that were necessary to sufficiently transmit the more strongly scattered probe light through the polycrystal. Due to the small size of the sample and the local nature of the pump and probe (both much smaller than the sample width), we have used a micro-diffraction setup (Bruker D8 Venture) to check for impurities or local decomposition. The trade-off for the spatial resolution of the measurement is the large instrument-limited peak broadening. Across more than 30 spots, no additional phases were detected beyond those previously identified (see Supplementary Figure S1 of Baclig *et al.*⁴⁸ for the full Rietveld fit) in the same commercial polycrystalline pellets, plus the inconel Ni-Cr alloy mounting plate. Representative results are now included in Supplementary Figure S4 and plotted below as Figure R5b. The variations in Figure R5b are attributable to the very small and slightly variable thickness of the sample, polished to translucence, which is comparable to the crystalline grain size. We rule out decomposition during TKE experiments.

Figure R5 | Stability of the TKE signal in Na β -alumina and post-measurement characterization. (a) TKE in a thin sample of Na β -alumina, measured four times (#1 through #4 chronologically) over ≈ 4 days with other measurements in between. There was a pause between the third and fourth measurements. Here, the peak pump field is applied at 1.5 ps. (b) X-ray microdiffraction characterization of the measured thin sample of K β'' -alumina. Color curves: seven micro-diffraction patterns collected from the polycrystalline K β'' -alumina sample following TKE measurements and additional 800nm illumination. The locations of major diffraction peaks plotted along the bottom are K β'' -alumina (black, ICSD 200993, with lattice parameters adjusted as in the fit reported by Baclig *et al.*⁴⁸), cubic and tetragonal zirconia (dark blue, ICSD 66781, and light blue, ICSD 26488, respectively), and Ni₃Cr alloy (grey, COD 1525114) to represent the Inconel support on which the sample is mounted. (c) Optical microscope images of all thin samples mounted for TKE measurements; scale bars are 200 μ m. (d) Optical microscope images of thicker β -alumina samples used for TKE measurements; scale bars are 1 mm.

Reviewer 2 (i): In a number of places, the references cited are difficult to reconcile with the statements made. I would encourage the authors to be more specific in their use of citations and comparison with prior literature. Also, in several places “agreement” with prior reported values (such as ion frequencies) is stated, but the reader is not provided with all the values to directly assess the level of “agreement”.

In response to the Reviewer’s request for additional references, we include Table R1 (added to the SI as Table S1) summarizing high-frequency NMR and conductivity measurements. The lowest-temperature regime of the NMR relaxation rates or GHz conductivities represents an upper bound

on the expected activation energies for higher-frequency methods such as ours by time-temperature superposition¹¹. In all samples, our TKE measurements yield activation energies below the NMR or GHz ones, which constitutes agreement with literature. The vibrational resonances in β -aluminas have also been measured extensively. The review by Lucazeau³⁴, cited in the main text, provides an exhaustive compendium. We will be interested in any specific references and comparisons that we may have missed, or any specific ones that the Reviewer would ask us to reconcile in more detail.

Table R1: high-frequency activation energies from NMR relaxation rates and high-frequency conductivity measurements (denoted as GHz). The lack of an uncertainty value for Ref. 16 reflects that the value is taken from Ref. 16 directly rather than from our own fitting, and uncertainty is not reported there. The lack of uncertainty values for Barker *et al.*¹⁷ reflects the lack of clarity on the co-variation of data reported there (see their Figure 5), or our use of their reported values. The lack of uncertainty values for Funke¹⁹ reflects the paucity of data points. Otherwise, our fitting uncertainty values are ± 1 s.e.

Material	Method	Frequency, MHz	Low T E_A , meV	High T E_A , meV	Ref.
Na β -alumina, melt	NMR	17.2	30 \pm 2	181 \pm 15	[13]
Na β -alumina, melt	NMR	25.5	40 \pm 2	105 \pm 4	[13]
Ag β -alumina	NMR	400	28 \pm 1	98 \pm 5	[14]
Na β -alumina, melt	NMR	21	42 \pm 2	135 \pm 4	[15]
Na β -alumina	NMR	5.2	39	N/A	[16]
K β -alumina	NMR	11.1	60 \pm 4	N/A	[5]
Na β -alumina, melt	GHz	1200	41	118	[17]
Na β -alumina, melt	GHz	24000	66		[17]
Ag β -alumina	GHz	380	24, 54	145	[17]
Ag β -alumina	GHz	24000	N/A	110	[17]
K β -alumina	GHz	1200	N/A	192	[17]
K β -alumina	GHz	24000	N/A	116	[17]
Na β'' -alumina	NMR	39.6	39 \pm 5, 98 \pm 7	252 \pm 9	[18]
Na β'' -alumina	GHz	18000	30	199	[19]
Na β'' -alumina	GHz	60000	29	N/A	[19]

Reviewer 3 writes (a): **I imagine that it might depend on the reader whether this paper is valuable or not. The significance of this work is also detailed in Sections 2 and 5.5 of the cited review paper (Ref 11).**

We thank the Reviewer for this comment. We also highlight that we isolate for the first time the attempt frequencies for ionic hopping, which constitute the fundamental vibrational origin of mass transport in the solid state. Until now, these have been “almost impossible to quantitatively measure experimentally” [Gao *et al.*, *Chem. Rev.* **120**, 5954 (2020)]. To our knowledge these have never been conclusively measured, yet they are crucial to understanding solid-state transport and thus designing transport properties for devices in energy technologies and neuromorphic computing. See also our comments above in reference to questions from reviewer 2.

Reviewer 3 (b): **In general, ionic conduction can be divided into two factors; enthalpy migration and entropy migration. Both factors have been investigated in superionic crystals including β -alumina from the viewpoint of physics for a long time, but only the enthalpy migration has been investigated with conventional methods by many researchers and developers of solid state ionics. Recently, the contribution of entropy migration has attracted attention again for improving solid electrolyte in novel fuel cells and all-solid-state batteries.**

We assume that the Reviewer means enthalpy of migration and entropy of migration. The entropy of migration remains key to practical predictions and materials design, because in its macroscopic interpretation it shifts Arrhenius conductivity-vs-inverse-temperature curves up or down for the same enthalpy (activation energy). However, to isolate the entropic component at the atomistic level has to our knowledge never been accomplished. We believe that the present study provides a new atomistic viewpoint on the subject.

Reviewer 3 (c): **THz spectroscopy and ultrafast time-resolved measurements were shown in Ref 11 as recent probing tools. THz Kerr effect measurement, widely used to investigate polar liquids, is based on both combinations. In this way, the authors succeeded in observing a phenomenon peculiar to entropy migration. The experimental results are also very reliable. In particular, the authors carefully checked the group velocity mismatch between the pump and probe pulse. ... However, I highly appreciate that their work is the first demonstration to evaluate ion dynamics in ionic conductors clearly on the picosecond time scale. Basically, I consider that this work deserves publication in Nature.**

We are grateful to the Reviewer for these encouraging comments. We hope that our work bringing in concepts of information theory can provide a novel perspective and lead to useful exploitation of these concepts, such as for accelerated predictions.

Reviewer 3 (d): **Unfortunately, the authors focused on the traditional superionic conductor of β -alumina, which might be considered by some readers as an increment of old experiments. I imagine that single crystallization is necessary in their demonstration to use near infrared probe pulses, which implies that their proposed technique might not be applicable to popular sintered solid electrolytes. The authors may suppress the appeal in this manuscript.**

We stress that the beta-aluminas were not investigated for novelty, but for the extensive body of literature on documenting their various properties as an exemplar model system, and in particular measuring their conductivities at a variety of frequencies.

We have used a carefully polished very thin polycrystalline sample of K β'' -alumina for the measurements shown in Figure 4 of the manuscript. In general, some transparency at probe wavelength, which is easiest to achieve with single crystals, or a specular reflection from a polished surface to measure in reflection mode, are necessary for a TKE measurement. It should therefore be possible to sinter, or hot-press and then polish, or fabricate with thin-film deposition techniques, a wide variety of solid-state ion-conducting materials for TKE measurements. We note that glassy ion conductors can also be made transparent in bulk.

The β -aluminas and β'' -alumina samples instead present a challenge with a relatively low concentration of mobile ions, which generally reduces the absolute magnitude of the signal. We have used them precisely of the large existing body of literature characterizing their vibrational and transport properties, which (a) has given us confidence that exclusively mobile-ion vibrations would be excited near resonance, and (b) has assisted in the interpretation of our novel signals and in eliminating the otherwise possible nonlinear-phononic or rotational contributions to birefringence. These are directly relevant to our unique experiments and novel concepts. For example, the absence of rotational modes and the trigonal lattice for ionic hopping in β -aluminas enable the detailed interpretation of experimental observables in our study.

Reviewer 3 (e): **However, I consider that the comparison between picosecond ionic motion and long-time motion is too rough. The authors have to measure AC impedance for their samples and have to compare it carefully. This is also important for showing their sample information. We see the ionic conductivity of Na β -alumina in many review papers, which shows the activation energy changing at 200°C. However, the authors estimated the activation energy from THz nonlinearity in a wide temperature range.**

It is the β'' -aluminas that exhibit two temperature regimes of activation, not the β -aluminas, which possess a single low-frequency activation energy at all practical temperatures. Therefore, our estimate of the TKE activation for K β -alumina (Figure 3b) stands across all temperatures and needs no further measurements. We agree that the field sometimes refers to these two families of materials interchangeably, but their conduction mechanisms are distinct.

The two β'' -alumina temperature regimes of activation (low activation at high temperature and higher activation at room temperature) are probed for a K β'' -alumina sample in Figure 4. We do not measure an activation energy for TKE, as we expect it to be small, and focus instead on the qualitative difference between two temperature regimes: the TKE tail is absent in the high-temperature regime due to the transport process becoming a random walk. We discuss this further in Ref. 7. Notably, to measure two activation regimes of conductivity in a β'' -alumina, a more transparent (or single-crystalline) sample would be needed. We do not possess one, otherwise we would have measured it.

Reviewer 3 (e): **In addition, the authors discuss the THz induced birefringence caused by ions hopping to adjacent sites, but I suspect whether terahertz electric fields really cause ions to**

hop to adjacent sites. In electronic systems, the kinetic energy accelerated by the half-cycle of the AC electric field is an important parameter for nonlinear optics. I roughly estimated that the kinetic energy of ion driven by an electric field of 300 kV/cm at 0.5 ps in the case of Ag (atomic mass of ~108) is $1/2 mv^2 = 2e^2 E^2 t^2 / m = 0.4 \text{ meV}$, which is much smaller than the estimated energy of potential barrier (several tens of meV). Intuitive explanation is needed that the observed transient dipole originates from ion hopping.

The energy deposited in the mobile ions by the terahertz pumps is indeed rather small. Overall, we simulate the pump-driven temperature rise to be on the order of 10 degrees Celsius for mobile ions only, and at most 1 degree Celsius when the energy is thermalized to the lattice (Supplementary Figures S6 and S7). This small heating is further consistent with small overall displacements in simulation (Supplementary Note 4): *“The simulated coherent displacements of the mobile-ion center of mass are between 5-15 picometers during the application of the pulse, in line with terahertz-frequency pump-probe studies in solid-state materials.”* [References: von Hoegen *et al.*⁴⁹, Kozina *et al.*³⁶, and Neugebauer *et al.*⁵⁰]

However, we eliminate possible phononic, nonlinear-phononic or rotational contributions to the measured TKE tails. We are not aware of any other contributions or processes that could yield the measured response, but we will consider specific ones that the Reviewer may suggest. The only remaining possible mechanism for a picosecond-timescale transient birefringence is the hopping of mobile ions, and our simulations verify that computationally. Our simulations also verify that hopping remains rare when the material is pumped with the THz pulses. Only a small minority of ions are hopping at any point in time (Boltzmann thermal distribution) and only a minority are driven to hop by the pump pulses, consistent with the Reviewer’s intuition. Driving a large fraction of ions to hop would result in the emergence of additional exotic intensity-dependent signals above some threshold excitation, which we also exclude in our experiment.

But the key conceptual advance in this study and in our Ref. 7 is that the overall energy corresponding to the macroscopic activation of conductivity (“tens of meV” or more) is not dissipated over the course of a single hopping event and does not have to correspond to a single-event atomistic energy barrier. Instead, except for the high-temperature regime in Figure 4, the full activation energy is dissipated over multiple sequential correlated hopping events and is furthermore a collective quantity averaged over multiple hopping pathways (see Ref. 7). The full macroscopic activation energy and entropy of conduction do not in general correspond to any singular atomistic event, as we argue in Ref. 7 and verify experimentally in the present manuscript.

In information-thermodynamics terms, the fastest hopping, probed here, is the most correlated (possessing the lowest information entropy) and therefore also corresponds to the lowest enthalpy (activation energy) by Landauer’s principle. We discuss this further in Supplementary Note 5. This allows for an activation of atomistic hopping that is much weaker than the macroscopic activation of overall conductivity or migration. As we have written in the main text, this information-theory picture and our results are consistent with the trend of decreasing measured activation energies with increasing frequency of measurement, which has been highlighted by Kamishima *et al.*¹² most clearly (see their Figure 6), and which we take to the fastest possible extreme in the present study. In fact, our combination of experiments and simulations enables us to explain this multiscale trend.

References in the present response document:

1. McWhan, D. B., Shapiro, S. M., Remeika, J. P. & Shirane, G. Neutron-scattering studies on beta-alumina. *J. Phys. C Solid State Phys.* **8**, L487 (1975).
2. Shapiro, S. M. & Reidinger, F. Neutron Scattering Studies of Superionic Conductors. in *Physics of Superionic Conductors* (ed. Salamon, M. B.) 45–75 (Springer-Verlag, 1979). doi:10.1007/978-3-642-81328-3_3.
3. Kadkhodaei, S. & Davariashdiyani, A. Phonon-assisted diffusion in bcc phase of titanium and zirconium from first principles. *Phys. Rev. Mater.* **4**, 043802 (2020).
4. Gupta, M. K. *et al.* Fast Na diffusion and anharmonic phonon dynamics in superionic Na₃PS₄. *Energy Environ. Sci.* **14**, 6554–6563 (2021).
5. Greenbaum, S. G. & Strom, U. Low-temperature nuclear spin relaxation in β -aluminas. *Solid State Commun.* **46**, 437–440 (1983).
6. Whittingham, M. S. & Huggins, R. A. BETA ALUMINA - PRELUDE TO A REVOLUTION IN SOLID STATE ELECTROCHEMISTRY. in *Proceedings of the 5th Materials Research Symposium* (eds. Roth, R. S. & Schneider S.J. Jr) 139–154 (National Bureau of Standards, 1972).
7. Radzilowski, R. H., Yao, Y. F. & Kummer, J. T. Dielectric loss of beta alumina and of ion-exchanged beta alumina. *J. Appl. Phys.* **40**, 4716–4725 (1969).
8. Funke, K., Cramer, C. & Wilmer, D. Concept of mismatch and relaxation for self-diffusion and conduction in ionic materials with disordered structures. in *Diffusion in Condensed Matter* (eds. Heitjans, P. & Kärger, J.) 857–893 (Springer-Verlag, 2005). doi:10.1007/3-540-30970-5_21.
9. Song, S. *et al.* Transport dynamics of complex fluids. *Proc. Natl. Acad. Sci.* **116**, 12733–12742 (2019).
10. Andersson, R., Årén, F., Franco, A. A. & Johansson, P. Ion Transport Mechanisms via Time-Dependent Local Structure and Dynamics in Highly Concentrated Electrolytes. *J. Electrochem. Soc.* **167**, 140537 (2020).
11. Dyre, J. C., Maass, P., Roling, B. & Sidebottom, D. L. Fundamental questions relating to ion conduction in disordered solids. *Reports Prog. Phys.* **72**, 046501 (2009).
12. Kamishima, O. *et al.* Temperature dependence of low-lying phonon dephasing by ultrafast spectroscopy (optical Kerr effect) in Ag β -alumina and Tl β -alumina. *J. Phys. Condens. Matter* **19**, 456215 (2007).
13. Walstedt, R. E., Dupree, R., Remeika, J. P. & Rodriguez, A. Na²³ nuclear relaxation in Na β -alumina: Barrier-height distributions and the diffusion process. *Phys. Rev. B* **15**, 3442–3454 (1977).
14. Iwai, Y., Kamishima, O., Kuwata, N., Kawamura, J. & Hattori, T. 109Ag NMR and relaxation mechanism in single crystal Ag β -alumina. *Solid State Ionics* **179**, 862–866 (2008).
15. Bjorkstam, J. L. & Villa, M. NMR studies of superionic β -aluminas. *J. Phys.* **42**, 345–351 (1981).
16. Greenbaum, S. G., Strom, U. & Rubinstein, M. NMR study of low-energy excitations in Na β -alumina. *Phys. Rev. B* **26**, 5226–5229 (1982).
17. Barker, A. S., Ditzenberger, J. A. & Remeika, J. P. Lattice vibrations and ion transport spectra in β -alumina. II. Microwave spectra. *Phys. Rev. B* **14**, 4254–4265 (1976).
18. Bjorkstam, J. L., Villa, M. & Farrington, G. C. Temperature dependence of the Na⁺ distribution in β -aluminas. *Solid State Ionics* **5**, 153–156 (1981).

19. Funke, K. Ion dynamics and correlations. *Philos. Mag. A* **68**, 711–724 (1993).
20. Deng, Z. *et al.* Fundamental investigations on the sodium-ion transport properties of mixed polyanion solid-state battery electrolytes. *Nat. Commun.* **13**, 4470 (2022).
21. He, X., Zhu, Y. & Mo, Y. Origin of fast ion diffusion in super-ionic conductors. *Nat. Commun.* **8**, 15893 (2017).
22. Gao, Y. *et al.* Classical and Emerging Characterization Techniques for Investigation of Ion Transport Mechanisms in Crystalline Fast Ionic Conductors. *Chem. Rev.* **120**, 5954–6008 (2020).
23. Cover, T. M. & Thomas, J. A. Maximum Entropy. in *Elements of Information Theory* 409–425 (Wiley, 2005). doi:10.1002/047174882X.ch12.
24. Klafter, J. & Sokolov, I. M. Continuous-time random walks. in *First Steps in Random Walks* 36–53 (Oxford University Press, 2011). doi:10.1093/acprof:oso/9780199234868.003.0003.
25. Meyer, M., Maass, P. & Bunde, A. A unified model for ion conduction in crystals of β - and β'' -alumina structure. *J. Chem. Phys.* **109**, 2316–2324 (1998).
26. Collin, D. *et al.* Verification of the Crooks fluctuation theorem and recovery of RNA folding free energies. *Nature* **437**, 231–234 (2005).
27. Bérut, A. *et al.* Experimental verification of Landauer’s principle linking information and thermodynamics. *Nature* **483**, 187–189 (2012).
28. Siria, A. *et al.* Giant osmotic energy conversion measured in a single transmembrane boron nitride nanotube. *Nature* **494**, 455–458 (2013).
29. Secchi, E. *et al.* Massive radius-dependent flow slippage in carbon nanotubes. *Nature* **537**, 210–213 (2016).
30. Mouterde, T. *et al.* Molecular streaming and its voltage control in ångström-scale channels. *Nature* **567**, 87–90 (2019).
31. Kavokine, N., Bocquet, M.-L. & Bocquet, L. Fluctuation-induced quantum friction in nanoscale water flows. *Nature* **602**, 84–90 (2022).
32. Yang, J. *et al.* Direct observation of ultrafast hydrogen bond strengthening in liquid water. *Nature* **596**, 531–535 (2021).
33. Mason, P. E. *et al.* Spectroscopic evidence for a gold-coloured metallic water solution. *Nature* **595**, 673–676 (2021).
34. Lucazeau, G. Infrared, Raman and neutron scattering studies of β - and β'' -alumina: a static and dynamical structure analysis. *Solid State Ionics* **8**, 1–25 (1983).
35. Först, M. *et al.* Nonlinear phononics as an ultrafast route to lattice control. *Nat. Phys.* **7**, 854–856 (2011).
36. Kozina, M. *et al.* Terahertz-driven phonon upconversion in SrTiO₃. *Nat. Phys.* **15**, 387–392 (2019).
37. Sajadi, M., Wolf, M. & Kampfrath, T. Terahertz-field-induced optical birefringence in common window and substrate materials. *Opt. Express* **23**, 28985 (2015).
38. Maehrlein, S. F. *et al.* Decoding ultrafast polarization responses in lead halide perovskites by the two-dimensional optical Kerr effect. *Proc. Natl. Acad. Sci.* **118**, e2022268118 (2021).
39. Huber, L., Maehrlein, S. F., Wang, F., Liu, Y. & Zhu, X. Y. The ultrafast Kerr effect in anisotropic and dispersive media. *J. Chem. Phys.* **154**, (2021).
40. Elgabarty, H. *et al.* Energy transfer within the hydrogen bonding network of water following resonant terahertz excitation. *Sci. Adv.* **6**, 1–15 (2020).

41. Zalden, P. *et al.* Molecular polarizability anisotropy of liquid water revealed by terahertz-induced transient orientation. *Nat. Commun.* **9**, 1–7 (2018).
42. Balos, V. *et al.* Time-resolved terahertz–Raman spectroscopy reveals that cations and anions distinctly modify intermolecular interactions of water. *Nat. Chem.* (2022) doi:10.1038/s41557-022-00977-2.
43. Kamishima, O., Iwai, Y., Kawamura, J. & Hattori, T. Defect modes around low-lying phonon in Ag β -Alumina by Raman scattering with high resolution. *Solid State Ionics* **179**, 780–782 (2008).
44. Kamishima, O., Iwai, Y., Hattori, T., Kawamura, K. & Kawamura, J. Vibrational analysis of ion dynamics in Ag β -alumina by Raman and molecular dynamics simulation. *J. Phys. Soc. Japan* **79**, 33–36 (2010).
45. Kamishima, O., Kawamura, K., Hattori, T. & Kawamura, J. Origin of activation energy in a superionic conductor. *J. Phys. Condens. Matter* **23**, (2011).
46. Kamishima, O., Iwai, Y. & Kawamura, J. Frequency dependence of ionic conductivity in a two-dimensional system of Ag β -alumina. *Solid State Ionics* **262**, 495–499 (2014).
47. Kamishima, O., Iwai, Y. & Kawamura, J. Small power-law dependence of ionic conductivity and diffusional dimensionality in β -alumina. *Solid State Ionics* **281**, 89–95 (2015).
48. Baclig, A. C. *et al.* High-Voltage, Room-Temperature Liquid Metal Flow Battery Enabled by Na-K|K- β'' -Alumina Stability. *Joule* **2**, 1287–1296 (2018).
49. Von Hoegen, A., Mankowsky, R., Fechner, M., Först, M. & Cavalleri, A. Probing the interatomic potential of solids with strong-field nonlinear phononics. *Nature* **555**, 79–82 (2018).
50. Neugebauer, M. J. *et al.* Comparison of coherent phonon generation by electronic and ionic Raman scattering in LaAlO₃. *Phys. Rev. Res.* **3**, 013126 (2021).

Reviewer Reports on the First Revision:

Referees' comments:

Referee #1 (Remarks to the Author):

A. Summary of the key results

The authors report on a nonlinear optical measurement to probe the ionic hopping in beta aluminas triggered by terahertz excitation pulse. The work further strengthens the understanding in the relation between vibrations and ionic conduction in solid state ionic conductors with a direct probe measurement tool. The measurement results are analyzed against simulations, which together show the memory in the non-ergodic system. This questions the nature of the random-walk and correlation between ionic jumps and transport.

B. Originality and significance: if not novel, please include reference

Original application of a method targeted onto predicted/simulated behavior of ionic jump. Significantly puts forward the discussion about the nature of ion transport due to generality with relevance into many fields and applications.

C. Data & methodology: validity of approach, quality of data, quality of presentation

Discussed in depth previously. All comments addressed in depth, really explained and extended highly competently.

Rev1 b) It is still a fascinating experimental setup results and careful analysis. In my opinion starting in the well-known model system is a strength of the study, I look forward to see the application to other systems and thereby the empirical evidence of generality of the concept especially when significantly altered host-lattices vibrational properties play a role.

Rev 1 c) A: This reproducibility and stability of the signal is indeed very assuring as is the absence of a drift.

D. Appropriate use of statistics and treatment of uncertainties

Previous comments addressed fully and considered carefully.

Rev1:d) Thank you for considering the comment carefully, I was wondering especially about the discrepancy and hope to have not misunderstood the actually different values from literature and your study. Giving an interval in that case with explanation is certainly helpful. I agree, considering the anharmonicity or somehow defining an anharmonicity to vibrational properties especially with moving ion is beyond the work.

Rev1: e) I see now, this was more about attributing an order of activation energy to more qualitatively confirm the expected range of it. This was not obvious to me from the manuscript. I think giving this value is still reasonable also when you rightfully so are only comfortable with giving it within one (type of) measurement. Estimating uncertainties is still valuable in my opinion for new methodology, or new application of measurement.

Rev1 f): Thank you for the clarification, I can understand that Supplementary Note 4 is considered the note relevant for that.

E. Conclusions: robustness, validity, reliability

Rev1 g) Thank you a lot for the extensive clarifications and the acknowledgement of a non-trivial answer to a comment. I hope that some of the clarifications made are something that helps the solid-state ionics community to understand the concept of the work presented. The additions made

to the supporting information are valuable.

With the given experiments and data which are robust the conclusions are valid within the tested system and the theoretical framework presented.

F. Suggested improvements: experiments, data for possible revision

Rev1 h) Thank you for acknowledging and explaining even very minor comments to the full extend.

No suggested improvements remain, publish as is.

G. References: appropriate credit to previous work?

Carefully considered in revision, even though the conceptual similarity

Rev1 i) I think their work shows, the relevancy of the specific modes contributing to (also longer range) ion (mass) transport. And basically by that also that that involving a phonon dispersion has to be considered going further. Which is, in my opinion, still not necessarily trivial and why I thought the reference may be helpful to include even with the pointed out differences. The reference is in my opinion still conceptually interesting and relevant addition to the manuscript.

H. Clarity and context: lucidity of abstract/summary, appropriateness of abstract, introduction and conclusions

Rev1 j) Thank you very much for extensive clarification, especially “whether each hop is Markovian versus whether the ergodic ensemble-average transport exhibits non-unity correlation factors when measured macroscopically”. Reducing fudge-factors means understanding those processes better, which are empirically investigated and are a valuable contribution to the fundamentals of ion transport. The incorporation into the introduction are helpful for the reader and a valuable improvement of manuscript.

Referee #2 (Remarks to the Author):

The authors provided an extensive and thorough rebuttal to the points brought up by all the reviewers. I initially had reservations about some aspects related to the novelty of the method and materials studied, however the authors developed strong arguments to convince me of the sufficient importance of the fundamental insights developed through the combination of TKE measurements and MD simulations of the THz pulse response. Further, it is important that the authors now more clearly explained the low magnitude of the THz pulse intensity on an atomistic scale. The study presents a new angle on the importance of vibrational dynamics and atomic correlations in the diffusion process. I would now be in favor of publishing in Nature this detailed study of fundamental mechanisms of THz-driven ionic hopping in the archetypal beta-aluminas.

However, I believe the manuscript would be great improved by including the figures and tables that the authors provided in the rebuttal. Some of these aspects may be provided if the review files accompany the published paper, but I still think the paper would be improved if these additional data and figures are included, at least in supplement (for instance figs R1-R4, sample characterization, table of activation energies, etc).

Referee #3 (Remarks to the Author):

The authors provide novel and complementary insights into the ion jump dynamics in the time domain by investigating the temporal evolution of THz-induced Kerr effect in this paper, which are experimentally inaccessible by other conventional techniques. This experimental technique is popular in other various materials and there are many relevant reports in high-IF journals. I agree that applying it to solid-state ion transport, an increasingly discussed field for energy storage applications, provides us fundamental insights.

These experimental results are evaluated through numerical calculations. In the last review round, I commented that this short-range transport should be recognized as an essential phenomenon as well as the long-range transport many researchers in solid-state ionics have focused on. Thus, I commended their study as the first demonstration to clearly evaluate the ion dynamics of ion conductors on a picosecond timescale.

However, I pointed out two unclear points in the previous round, which are interconnected. One question is how microscale ion dynamics contribute to long-range ion diffusion. Ion hopping between two sites is also observed in ferroelectrics, where the connection to long-distance ion conduction is inherently unclear. Another question is the discrepancy between the intuitive understanding of proton hopping despite the very low electric field intensity of the pump light. These points highlight the possibility that the observed THz Kerr signal may arise from other factors. The authors comment on the exclusion of possibilities such as nonlinear phononics or contributions from rotations, which are reasonable considerations.

I believe that it is deserving of publication in Nature as it is.

Author Rebuttals to First Revision:

We are grateful to the editor and to the reviewers for their time and their positive comments. The peer review process has helped us to sharpen the text and our thoughts. We intend to opt into transparent peer review for this next submission. Here we respond to the reviewers' comments point by point. Changes to the manuscript or supplementary information are highlighted in yellow.

Referee #1: A. Summary of the key results

The authors report on a nonlinear optical measurement to probe the ionic hopping in beta aluminas triggered by terahertz excitation pulse. The work further strengthens the understanding in the relation between vibrations and ionic conduction in solid state ionic conductors with a direct probe measurement tool. The measurement results are analyzed against simulations, which together show the memory in the non-ergodic system. This questions the nature of the random-walk and correlation between ionic jumps and transport.

B. Originality and significance: if not novel, please include reference

Original application of a method targeted onto predicted/simulated behavior of ionic jump. Significantly puts forward the discussion about the nature of ion transport due to generality with relevance into many fields and applications.

C. Data & methodology: validity of approach, quality of data, quality of presentation. Discussed in depth previously. All comments addressed in depth, really explained and extended highly competently.

We are very grateful to the Reviewer for their in-depth questions and positive comments. We hope this study can contribute to our collective understanding of transport in condensed phases.

Rev1 b) It is still a fascinating experimental setup results and careful analysis. In my opinion starting in the well-known model system is a strength of the study, I look forward to see the application to other systems and thereby the empirical evidence of generality of the concept especially when significantly altered host-lattices vibrational properties play a role.

We hope to carry out further measurements on more modern systems. We hope our study motivates a closer correspondence between fast-timescale experimental characterization and atomistic simulation.

Rev 1 c) A: This reproducibility and stability of the signal is indeed very assuring as is the absence of a drift.

We have included the raw experimental data for this and other runs of measurement in the open-source repository at [doi:10.5281/zenodo.8169682](https://doi.org/10.5281/zenodo.8169682).

D. Appropriate use of statistics and treatment of uncertainties. Previous comments addressed fully and considered carefully.

Rev1: d) Thank you for considering the comment carefully, I was wondering especially about the discrepancy and hope to have not misunderstood the actually different values from literature and your study. Giving an interval in that case with explanation is certainly helpful. I

agree, considering the anharmonicity or somehow defining an anharmonicity to vibrational properties especially with moving ion is beyond the work.

Rev1: e) I see now, this was more about attributing an order of activation energy to more qualitatively confirm the expected range of it. This was not obvious to me from the manuscript. I think giving this value is still reasonable also when you rightfully so are only comfortable with giving it within one (type of) measurement. Estimating uncertainties is still valuable in my opinion for new methodology, or new application of measurement.

For reproducibility and transparency, we are including additional detail on our fitting of TKE relaxation to exponentials and the assumptions involved with the online data accompanying this manuscript. These relate to (1) the shape of the rise of the ps-timescale non-oscillatory signal and (2) the treatment of long-time background.

We have made a small correction to the fitting of lifetimes – the earlier figure we cited was based on using an error function rise simplified from the field shape. We report the activation energy as 40.4 ± 4.8 meV, with a slightly smaller uncertainty from the previous estimate of 40.6 ± 6.8 meV which we had reported as 41 ± 7 meV. This is reflected in Figure 3b and in the methods: “A free constant is included as a long-time asymptote due to the finite time delays probed. This is consistent with wide distributions of hopping lifetimes [Ref: Poletayev *et al.*, *Nature Materials* (2022)]”. We include both the raw experimental data and the processing and fitting scripts at doi:10.5281/zenodo.8169682.

E. Conclusions: robustness, validity, reliability

Rev1 g) Thank you a lot for the extensive clarifications and the acknowledgement of a non-trivial answer to a comment. I hope that some of the clarifications made are something that helps the solid-state ionics community to understand the concept of the work presented. The additions made to the supporting information are valuable.

With the given experiments and data which are robust the conclusions are valid within the tested system and the theoretical framework presented.

F. Suggested improvements: experiments, data for possible revision

Rev1 h) Thank you for acknowledging and explaining even very minor comments to the full extend. No suggested improvements remain, publish as is.

We are very grateful to the Reviewer for their time and detailed comments that strengthened our manuscript.

G. References: appropriate credit to previous work?

Carefully considered in revision, even though the conceptual similarity

Rev1 i) I think their work shows, the relevancy of the specific modes contributing to (also longer range) ion (mass) transport. And basically by that also that that involving a phonon dispersion has to be considered going further. Which is, in my opinion, still not necessarily trivial and why I thought the reference may be helpful to include even with the pointed out differences. The reference is in my opinion still conceptually interesting and relevant addition to the manuscript.

We agree that considering a dispersion relation which resolves normal modes by their momentum will be a helpful next step to investigate the vibrational contributions to transport. We have noted in the Supplementary Information that for β/β'' -aluminas our results on the direct excitation of (directionally) non-thermal hopping offer a means to test various models both experimentally and with effectively digital-twin simulations.

H. Clarity and context: lucidity of abstract/summary, appropriateness of abstract, introduction and conclusions.

Rev1 j) Thank you very much for extensive clarification, especially “whether each hop is Markovian versus whether the ergodic ensemble-average transport exhibits non-unity correlation factors when measured macroscopically”. Reducing fudge-factors means understanding those processes better, which are empirically investigated and are a valuable contribution to the fundamentals of ion transport. The incorporation into the introduction are helpful for the reader and a valuable improvement of manuscript.

We are very grateful to the Reviewer for the opportunity to address the historic conceptual frameworks within which our fields consider correlations in ion transport. The book by H. Mehrer was a helpful read especially in tracking the use of the correlation factors to Bardeen. The Reviewer’s comments have been very helpful to our thinking. We hope to test the assumptions of Markovianity in relation to condensed-phase transport further in subsequent studies.

Referee #2 (Remarks to the Author):

The authors provided an extensive and thorough rebuttal to the points brought up by all the reviewers. I initially had reservations about some aspects related to the novelty of the method and materials studied, however the authors developed strong arguments to convince me of the sufficient importance of the fundamental insights developed through the combination of TKE measurements and MD simulations of the THz pulse response. Further, it is important that the authors now more clearly explained the low magnitude of the THz pulse intensity on an atomistic scale. The study presents a new angle on the importance of vibrational dynamics and atomic correlations in the diffusion process. I would now be in favor of publishing in Nature this detailed study of fundamental mechanisms of THz-driven ionic hopping in the archetypal beta-aluminas.

However, I believe the manuscript would be great improved by including the figures and tables that the authors provided in the rebuttal. Some of these aspects may be provided if the review files accompany the published paper, but I still think the paper would be improved if these additional data and figures are included, at least in supplement (for instance figs R1-R4, sample characterization, table of activation energies, etc).

We thank the Reviewer for their detailed reading of the manuscript and their encouraging comments. Figures R1-R5 are now included as Figures S4ef, S10, 3b, S12, and S4b-d, respectively.

Referee #3 (Remarks to the Author):

The authors provide novel and complementary insights into the ion jump dynamics in the time domain by investigating the temporal evolution of THz-induced Kerr effect in this paper, which

are experimentally inaccessible by other conventional techniques. This experimental technique is popular in other various materials and there are many relevant reports in high-IF journals. I agree that applying it to solid-state ion transport, an increasingly discussed field for energy storage applications, provides us fundamental insights.

These experimental results are evaluated through numerical calculations. In the last review round, I commented that this short-range transport should be recognized as an essential phenomenon as well as the long-range transport many researchers in solid-state ionics have focused on. Thus, I commended their study as the first demonstration to clearly evaluate the ion dynamics of ion conductors on a picosecond timescale.

However, I pointed out two unclear points in the previous round, which are interconnected. One question is how microscale ion dynamics contribute to long-range ion diffusion. Ion hopping between two sites is also observed in ferroelectrics, where the connection to long-distance ion conduction is inherently unclear. Another question is the discrepancy between the intuitive understanding of proton hopping despite the very low electric field intensity of the pump light. These points highlight the possibility that the observed THz Kerr signal may arise from other factors. The authors comment on the exclusion of possibilities such as nonlinear phononics or contributions from rotations, which are reasonable considerations.

I believe that it is deserving of publication in Nature as it is.

We are grateful to the reviewer for their encouraging comments in this and the previous round of peer review. We agree that the transient birefringence method we employ is now used to probe a variety of condensed and gaseous systems. We hope our references give credit to that breadth since the present work is informed by studies on all states of matter: liquids, gases, and solids. However, we believe ours is the first to attempt to characterize ion transport with this tool. Regarding ion hopping in ferroelectrics, we are unfortunately not aware of any studies triggering ion translation in such systems. We would like to have cited what we believe is a relevant study by Kozina *et al.* focusing on the soft mode shifting the orientation of SrTiO₃ (supplementary ref. 3). Our study does not involve proton hopping, although further studies of transport in liquids and in super-protonic conductors would be very interesting.